# Visual Diversity and Region-aware Prompt Learning for Zero-shot HOI Detection

**Chanhyeong Yang**[1]    **Taehoon Song**[2]    **Jihwan Park**[1]    **Hyunwoo J. Kim**[2*]

[1]Korea University    [2]Korea Advanced Institute of Science and Technology

{0814gerrardso, jseven7071}@korea.ac.kr {taehoons, hyunwoojkim}@kaist.ac.kr

## Abstract

Zero-shot Human-Object Interaction detection aims to localize humans and objects in an image and recognize their interaction, even when specific verb-object pairs are unseen during training. Recent works have shown promising results using prompt learning with pretrained vision-language models such as CLIP, which align natural language prompts with visual features in a shared embedding space. However, existing approaches still fail to handle the *visual complexity of interaction*—including (1) *intra-class visual diversity*, where instances of the same verb appear in diverse poses and contexts, and (2) *inter-class visual entanglement*, where distinct verbs yield visually similar patterns. To address these challenges, we propose **VDRP**, a framework for *Visual Diversity and Region-aware Prompt learning*. First, we introduce a visual diversity-aware prompt learning strategy that injects group-wise visual variance into the context embedding. We further apply Gaussian perturbation to encourage the prompts to capture diverse visual variations of a verb. Second, we retrieve region-specific concepts from the human, object, and union regions. These are used to augment the diversity-aware prompt embeddings, yielding region-aware prompts that enhance verb-level discrimination. Experiments on the HICO-DET benchmark demonstrate that our method achieves state-of-the-art performance under four zero-shot evaluation settings, effectively addressing both intra-class diversity and inter-class visual entanglement. Code is available at https://github.com/mlvlab/VDRP.

## 1   Introduction

Human-Object Interaction (HOI) detection [1, 2, 3, 4, 5, 6, 7, 8, 9, 10, 11, 12, 13] aims to localize humans and objects in an image and recognize the interactions between them, serving as a cornerstone for fine-grained scene understanding. Unlike standard HOI detection, which assumes supervision over all interactions, zero-shot HOI detection must generalize to unseen combinations of verbs and objects. While recent advances in Vision-Language Models (VLMs) have significantly improved the zero-shot recognition of object and attribute classes [14, 15, 16, 17, 18, 19, 20], zero-shot HOI detection remains fundamentally more challenging. As previous works [21, 22] pointed out, the difficulty stems not only from the compositional novelty of interactions but also from the visual complexity of interactions—where each verb class exhibits large intra-class visual diversity, and different verbs frequently produce visually similar patterns.

First, verb classes exhibit substantial intra-class diversity. As shown in Fig. 1-(A), instances of the verb "holding a baseball glove" may appear in drastically different poses, scales, or scene contexts, yet must be classified under the same label. To quantify this, we compute a diversity score using CLS features extracted from a frozen CLIP visual encoder. For verbs, we crop the union region; for objects, the object bounding box is used. The average pairwise cosine similarity within each class is

---

*Corresponding author

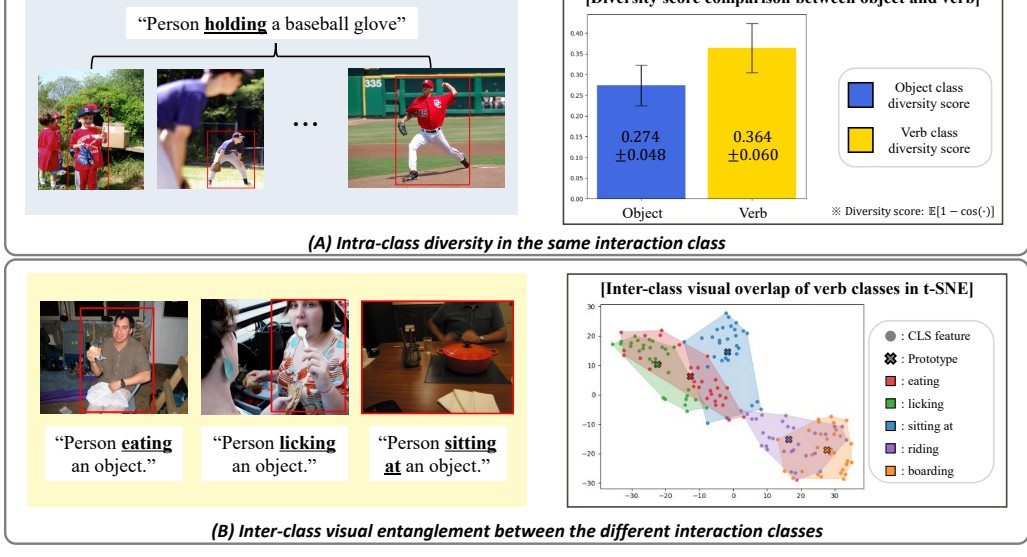

Figure 1: **Analysis of the visual complexity in HOI detection.** (A) Verb classes exhibit significant *intra-class visual diversity*, where instances of the same verb (e.g., "holding a baseball glove") appear under varied poses, viewpoints, and scene contexts. To quantify this, we crop the union region and extract the CLIP visual CLS feature. A diversity score is then computed as the expected cosine dissimilarity $\mathbb{E}[1 - \cos(\cdot)]$ across samples of the same class. Verb classes exhibit higher diversity $(0.364 \pm 0.060)$ than object classes $(0.274 \pm 0.048)$, highlighting the difficulty of representing verbs with a single static embedding. (B) Verb classification also suffers from *inter-class visual entanglement*, where semantically distinct verbs (e.g., "eating", "licking", "sitting at") yield visually similar patterns. To visualize this, we randomly select five verb classes, extract their union-region CLS features, and project them to 2D using t-SNE. The resulting clusters show significant overlap, highlighting the need for region-aware prompts to improve verb separability in HOI detection.

measured, and the diversity score is defined as the expectation of $1 - \cos(\cdot)$. This analysis shows that verbs have significantly higher intra-class diversity $(0.364 \pm 0.060)$ compared to objects $(0.274 \pm 0.048)$, indicating that a single prompt embedding is likely insufficient to capture such variation.

Second, interactions often exhibit *inter-class visual entanglement*, where different verbs produce highly similar visual patterns. As illustrated in Fig. 1-(B), semantically distinct verbs such as "eating", "licking", and "sitting at" frequently share similar human-object layouts. In such cases, accurate classification depends on regional differences—often localized in the human or object region—which global or union-level features may fail to capture. To analyze this phenomenon, we extract union-region CLS features from a frozen CLIP visual encoder across five verbs and project them into a 2D space using t-SNE. The resulting convex hulls show substantial overlap not only at the sample level but also across verb prototypes, revealing poor inter-class separability. This entanglement poses a major challenge for zero-shot HOI detection, where explicit class supervision is unavailable.

Recently, studies on zero-shot HOI detection have explored prompting strategies that leverage pretrained CLIP models [23, 24, 25, 26, 27]. These methods map HOI triplets to textual descriptions and embed them in a shared vision-language space. While effective for semantic alignment, most approaches assume a single static prompt per verb [23, 10, 26], making them inadequate for modeling the visual diversity within each class. Some incorporate spatial cues in the visual branch [24], but leave the text prompts agnostic to region-specific semantics. Others rely on LLM-generated descriptions [25], focusing on semantic differences across verbs but overlooking intra-class variation.

To overcome these limitations, we propose a new framework called **VDRP** (**V**isual **D**iversity and **R**egion-aware **P**rompt learning) for zero-shot HOI detection. Our method enhances prompt representations in two complementary ways. First, we inject group-wise visual variance into the learnable context embeddings and apply Gaussian perturbation, allowing the prompts to reflect intra-class appearance diversity and better adapt to varied visual realizations during training. Second, we retrieve

region-specific concepts from the human, object, and union regions and use them to augment the prompt embeddings, producing region-aware prompts that improve verb discriminability.

Our **contributions** are summarized as follows:

- We propose a *visual diversity-aware prompt learning* that models intra-class variation by injecting group-wise variance into the context embedding and applying Gaussian perturbation, enabling the prompts to generalize across diverse appearances of the same verb.

- We introduce a *region-aware prompt augmentation* that leverages region-specific concept retrieval from human, object, and union regions to enrich the prompts, enhancing discriminability among visually similar verbs.

- We integrate the two modules into a unified framework, **VDRP**, and demonstrate its effectiveness on the HICO-DET benchmark, achieving new state-of-the-art performance under multiple zero-shot evaluation settings.

## 2 Related works

**HOI detection.** Human-Object Interaction (HOI) detection typically involves three sub-tasks: object detection, human-object pairing, and interaction classification. Thanks to advances in large-scale benchmarks [28, 29, 4, 21] and transformer-based architectures [30, 9, 31], a wide range of approaches have been proposed. Broadly, HOI methods fall into one-stage [23, 27, 4, 9, 11, 7] and two-stage [10, 26, 24, 25, 5, 6] paradigms. One-stage methods jointly predict object locations and interactions, often using set prediction frameworks such as DETR [30]. In contrast, two-stage methods decouple the task into object detection and interaction classification: a pre-trained detector localizes humans and objects, and a dedicated module predicts the verb label for each human-object pair. The division of the HOI task in two-stage approaches allows efficient training [5, 6], and shows promising results. Our work falls in the two-stage paradigm.

**Zero-shot HOI detection.** Zero-shot HOI detection aims to identify HOI triplets unseen during training, a task challenged by the long-tail distribution of compositional datasets. With the rise of Vision-Language Models (VLMs) [15, 32, 33, 14, 34] pretrained on large-scale image-text pairs, many works leverage their generalization for HOI. Several methods [10, 23, 27, 35] align HOI models with CLIP's pretrained representations to enable effective transfer. Recent approaches [24, 26, 25] adopt prompt learning to adapt CLIP with few learnable parameters for fine-grained interaction understanding. However, most rely on a single static prompt per verb [10, 23, 26], limiting their ability to capture intra-class visual diversity. Later works add spatial cues or LLM-generated descriptions [24, 25], but still lack region-level adaptation or concept-level grounding. We address these gaps with visual diversity-aware prompt learning and region-aware prompt augmentation.

**Prompt learning.** Prompt learning is a widely used strategy for adapting vision-language models (VLMs) like CLIP to downstream tasks [36, 37, 38, 39, 40, 41, 42]. Early works such as CoOp [18] and CoCoOp [19] optimize learnable or image-conditioned context vectors, while MaPLe [38] jointly tunes visual and textual prompts to improve cross-modal alignment. In zero-shot HOI detection, CMMP [24] and EZ-HOI [25] adopt multi-modal prompt learning, but do not account for visual diversity of verbs and use the same verb prompt across regions. This limits their ability to adapt to diverse verb appearances and handle visually similar interactions. Meanwhile, distribution-based prompt learning [37, 36, 40, 43] has shown that leveraging feature variance improves generalization. Inspired by this, we propose a method that injects group-wise visual variance—via modulation and perturbation—into prompt embeddings, and augments them with region-specific concepts, enabling effective handling of visual diversity and improving verb discriminability in zero-shot HOI detection.

## 3 Methods

In this section, we present our framework, **VDRP**, designed to address two key challenges in zero-shot Human-Object Interaction (HOI) detection: (1) *intra-class visual diversity*, where instances of the same verb exhibit a wide range of visual appearances, and (2) *inter-class visual entanglement*, where semantically distinct verbs appear visually similar. To tackle these challenges, VDRP introduces two complementary components. The first, *visual diversity-aware prompt learning*, addresses intra-class diversity by injecting group-wise visual variance into the context embeddings and applying variance-guided perturbation, resulting in *visual diversity-aware prompts* that better capture verb-specific

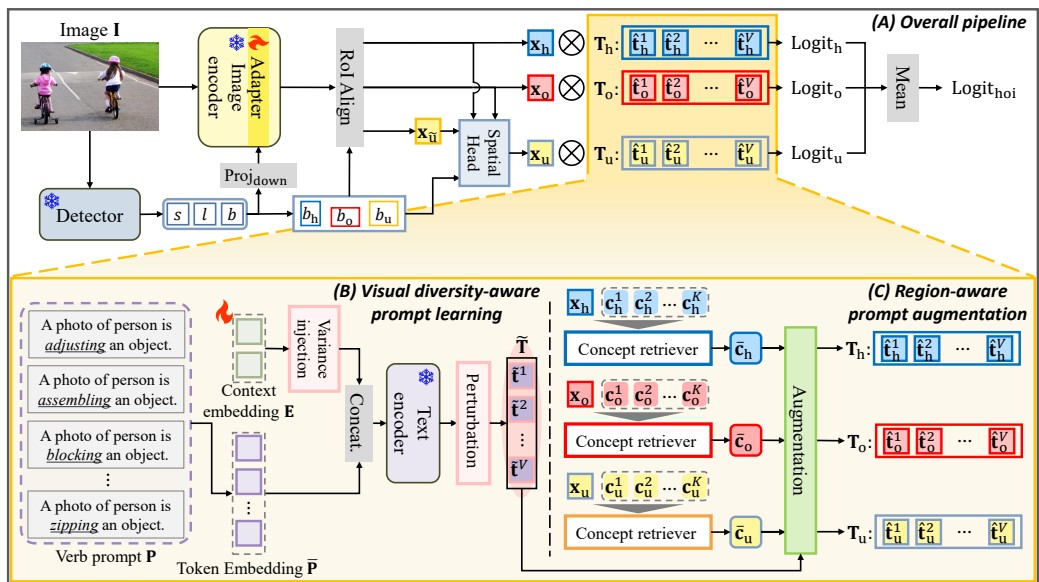

Figure 2: **Overview of our VDRP framework.** (A) We adopt a two-stage HOI detection pipeline with a frozen detector and a CLIP image encoder to extract human ($\mathbf{x}_h$), object ($\mathbf{x}_o$), and union ($\mathbf{x}_{\bar{u}}$) features. A spatial head further refines the union feature into $\mathbf{x}_u$ for region-aware prompts via spatial encoding. (B) Visual diversity-aware prompts are generated by injecting group-wise variance and perturbation to model intra-class variation. (C) Retrieved region concepts are then fused with these prompts to produce final region-aware prompts $\mathbf{T}_h$, $\mathbf{T}_o$, and $\mathbf{T}_u$ used for verb classification.

appearance variations. The second, *region-aware prompt augmentation*, enhances these prompts using region-specific concepts retrieved from the human, object, and union regions, yielding *region prompts* that improve inter-class discriminability. We begin by outlining the overall pipeline (Section 3.1), followed by detailed descriptions of the two modules in Sections 3.2 and 3.3.

## 3.1 Overall pipeline

As illustrated in Fig. 2-(A), our method follows a two-stage HOI detection framework [10, 24, 5, 6], consisting of (1) human-object detection and (2) interaction classification. In the first stage, a frozen object detector (DETR [30]) is applied to the input image $\mathbf{I}$ to identify object instances. Each detection yields a triplet $(s_n, l_n, b_n)$, representing the confidence score, object class embedding, and bounding box of the $n$-th instance. In the second stage, we encode each instance into a prior embedding $\mathbf{p} = \mathrm{Proj}_{\mathrm{down}}([s; l; b]) \in \mathbb{R}^{d_{\mathrm{down}}}$, which serves as guidance for task adaptation. Using this prior, we extract patch embeddings from a frozen CLIP encoder augmented with lightweight adapter layers inserted into multiple transformer blocks. Each adapter applies a bottleneck structure that projects the patch features, attends to the prior $\mathbf{p}$, and reconstructs the output:

$$X_i' = \mathbf{CrossAttn}(\mathrm{Proj}_{\mathrm{down}}(X_i), \mathbf{p}, \mathbf{p}),$$
$$X_{i+1} = X_i + \mathrm{Proj}_{\mathrm{up}}(X_i'), \tag{1}$$

where $X_i \in \mathbb{R}^{N \times d_{\mathrm{up}}}$ is the image feature map at layer $i$. The final image feature map $\mathbf{X} \in \mathbb{R}^{N \times d}$ is obtained from the last transformer layer with projection. Given the image features and the boxes for the human ($b_h$), object ($b_o$), and union region ($b_u$), we extract region features via RoIAlign [44]:

$$\mathbf{x}_h = \mathrm{RoIAlign}(\mathbf{X}, b_h), \quad \mathbf{x}_o = \mathrm{RoIAlign}(\mathbf{X}, b_o), \quad \mathbf{x}_{\bar{u}} = \mathrm{RoIAlign}(\mathbf{X}, b_u), \tag{2}$$

where $\mathbf{x}_h, \mathbf{x}_o, \mathbf{x}_{\bar{u}} \in \mathbb{R}^d$ denote the region features for the human, object, and union regions, respectively. To enhance the union-region representation with spatial priors, we apply a spatial head that fuses $\mathbf{x}_{\bar{u}}$ with human and object features, as well as their bounding boxes:

$$\mathbf{x}_u = \mathrm{SpatialHead}(\mathbf{x}_{\bar{u}}; \mathbf{x}_h, \mathbf{x}_o; b_h, b_o) \in \mathbb{R}^d. \tag{3}$$

More details on SpatialHead($\cdot$) are provided in the supplementary material. Next, each region feature—$\mathbf{x}_h$, $\mathbf{x}_o$, and $\mathbf{x}_u$—is matched against a set of verb prompts.

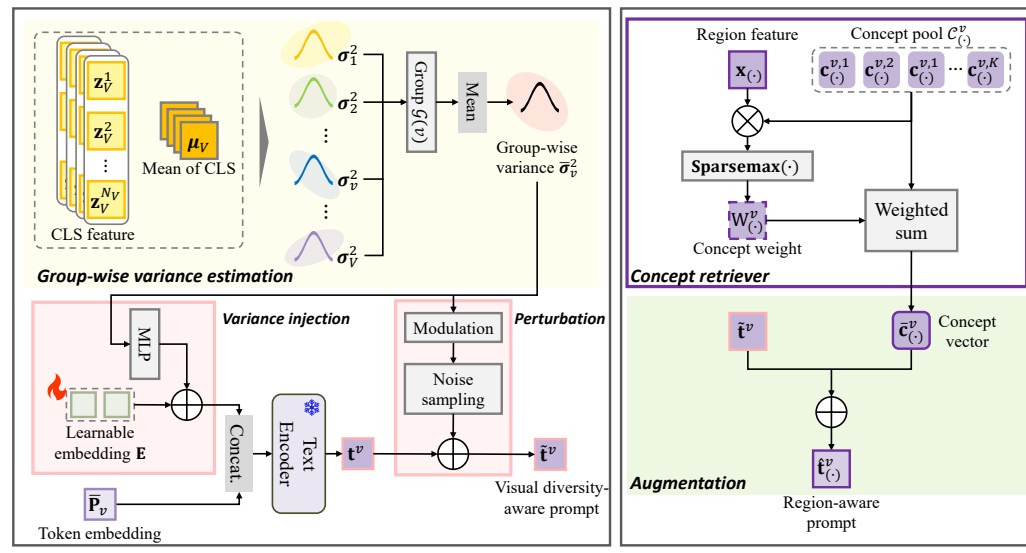

**(A) Visual diversity-aware prompt learning**  **(B) Region-aware prompt augmentation**

Figure 3: **Detailed architecture of our methods.** (A) To model intra-class variation, we compute verb-wise visual variance $\sigma_v^2$ from union-region CLS features, average them over similar verbs to obtain group-wise variance $\bar{\sigma}_v^2$, and inject it into the shared context embedding $\mathbf{E}$ via an MLP. This is combined with the verb prompt $\bar{\mathbf{P}}_v$ and encoded by the CLIP text encoder to produce $\mathbf{t}^v$, which is further perturbed using Gaussian noise scaled by visual variance. (B) For inter-class discriminability, we retrieve region concepts from features $\mathbf{x}_{(\cdot)}$ using a Sparsemax over a concept pool $\mathcal{C}_{(\cdot)}^v$, and add the result to $\tilde{\mathbf{t}}^v$ to obtain the final region-aware prompt $\hat{\mathbf{t}}_{(\cdot)}^v$.

As illustrated in Fig.2-(B) and (C), Each prompt $\hat{\mathbf{t}}_{(\cdot)}^v$ is constructed in two stages. Let $\mathbf{T}_{(\cdot)} = [\hat{\mathbf{t}}_{(\cdot)}^1, \ldots, \hat{\mathbf{t}}_{(\cdot)}^V] \in \mathbb{R}^{d \times V}$ denote the region-aware prompts for $(\cdot) \in \{\text{h}, \text{o}, \text{u}\}$. We first generate a visual diversity-aware prompt $\tilde{\mathbf{t}}^v$ via group-wise variance injection and Gaussian perturbation, then augment it with region-specific concepts retrieved based on region feature $\mathbf{x}_{(\cdot)}$. This two-step design allows the final region-aware prompts to reflect both the visual diversity of verbs and localized context. See Sections 3.2 and 3.3 for details. The logits for each region prompts is computed as:

$$\text{Logit}_\text{h} = \mathbf{x}_\text{h}^\top \mathbf{T}_\text{h}, \quad \text{Logit}_\text{o} = \mathbf{x}_\text{o}^\top \mathbf{T}_\text{o}, \quad \text{Logit}_\text{u} = \mathbf{x}_\text{u}^\top \mathbf{T}_\text{u}, \tag{4}$$

where $\text{Logit}_{(\cdot)} \in \mathbb{R}^V$. Finally, the overall HOI classification logit is obtained by averaging region-wise logits:

$$\text{Logit}_\text{hoi} = \frac{1}{3} \left( \text{Logit}_\text{h} + \text{Logit}_\text{o} + \text{Logit}_\text{u} \right). \tag{5}$$

We train the model using focal loss [45] for multi-label verb classification.

## 3.2 Visual diversity-aware prompt learning

To address intra-class visual diversity in HOI detection, we propose a visual diversity-aware prompt learning method that incorporates visual variance into both the context modulation and prompt perturbation processes. Static prompts—optimized as single points—struggle to represent the wide variation of instances within the same verb class. Inspired by recent findings [40, 37, 36] showing that variance-aware representations improve generalization, we explicitly encode class-level visual variance to guide prompt adaptation and inject noise that reflects the extent of such diversity.

**Group-wise variance estimation.** As illustrated in Fig. 3-(A), we begin by extracting union-region features from the training set using a frozen CLIP image encoder. Specifically, we crop the union box of each human-object pair and extract the CLS token. For each verb $v$, let $\mathbf{z}_v^{(j)}$ denote the CLS feature of the $j$-th instance. We compute the mean $\boldsymbol{\mu}_v$ and variance $\boldsymbol{\sigma}_v^2$ over all $N_v$ samples. To obtain a stable estimate for rare or unseen verbs, we construct a group of verbs $\mathcal{G}(v)$ by selecting the

similar verbs based on cosine similarity between CLIP text embeddings. This grouping allows each verb to inherit variance statistics from its semantically similar neighbors. The group-wise variance is then computed as:

$$\bar{\boldsymbol{\sigma}}_v^2 = \frac{1}{|\mathcal{G}(v)|} \sum_{v' \in \mathcal{G}(v)} \boldsymbol{\sigma}_{v'}^2. \tag{6}$$

This group-wise variance serves as an inductive prior capturing the expected diversity of verb $v$.

**Visual diversity-aware prompts.** Then, we transform the group-wise variance into a modulation vector using a lightweight MLP to perform variance injection as follows:

$$\mathbf{d}_v = \text{MLP}(\bar{\boldsymbol{\sigma}}_v^2) \in \mathbb{R}^d. \tag{7}$$

This is added to the shared context embedding $\mathbf{E} \in \mathbb{R}^{N_{\text{ctx}} \times d}$ to produce a verb-specific context:

$$\hat{\mathbf{E}}_v = \mathbf{E} + \mathbf{d}_v\, \alpha, \tag{8}$$

where $\alpha$ is a small scaling factor for stability. Given a verb prompt sentence $\mathbf{P}_v$ (e.g., "A photo of a person is $[v]$+ing an object."), we tokenize it into token embedding $\bar{\mathbf{P}}_v$ and concatenate it with the modulated context to form the final input for CLIP text encoder $\mathcal{T}(\cdot)$:

$$\mathbf{t}^v = \mathcal{T}([\hat{\mathbf{E}}_v; \bar{\mathbf{P}}_v]) \in \mathbb{R}^d. \tag{9}$$

To further reflect visual variability, we perturb each prompt embedding using Gaussian noise scaled by the group-wise standard deviation. We normalize $\bar{\boldsymbol{\sigma}}_v$ across dimensions and modulate it to match the standard deviation of $\mathbf{t}^v$, producing $\tilde{\boldsymbol{\sigma}}_v$. Noise is then sampled and applied element-wise:

$$\tilde{\mathbf{t}}^v = \mathbf{t}^v + (\boldsymbol{\epsilon} \odot \tilde{\boldsymbol{\sigma}}_v)\, \beta, \quad \boldsymbol{\epsilon} \sim \mathcal{N}(\mathbf{0}, \mathbf{I}), \tag{10}$$

where $\beta$ is a small scaling factor controlling the perturbation strength. This results in prompt embeddings that encode both the central semantics of the verb and its expected visual diversity. Finally, we collect all perturbed prompts across the $V$ verbs to form the diversity-aware prompts $\tilde{\mathbf{T}} = [\tilde{\mathbf{t}}^1, \ldots, \tilde{\mathbf{t}}^V] \in \mathbb{R}^{d \times V}$, which serves as the base for region-aware prompts.

### 3.3 Region-aware prompt augmentation

To address inter-class visual entanglement—where semantically distinct verbs exhibit similar visual patterns—we augment prompts using region-specific concepts from the human, object, and union regions. While diversity-aware prompts reflect class-level variation, they cannot capture the region concepts needed to distinguish visually similar verbs. To bridge this gap, we introduce retrieval-based region-aware prompt augmentation, which allows each prompt to specialize based on regions.

**Region concept generation.** To enrich prompt embeddings with localized semantics, we follow [46, 47, 20] and query LLMs (e.g., LLaMA-7B [48] and ChatGPT-4 [49]) to generate $K$ region-level concepts for each verb $v$ and region type $\mathbf{R} \in \{\text{human, object, union}\}$. Each prompt follows the format: "For the verb $[\mathbf{P}_v]$, give $K$ short visual concepts for the $[\mathbf{R}]$ region." The resulting concepts are encoded using the CLIP text encoder $\mathcal{T}(\cdot)$ to form the concept pool:

$$\mathcal{C}_{(\cdot)}^v = \left\{ \mathbf{c}_{(\cdot)}^{v,1}, \ldots, \mathbf{c}_{(\cdot)}^{v,K} \right\}, \quad \mathbf{c}_{(\cdot)}^{v,k} \in \mathbb{R}^d,$$

where $(\cdot) \in \{\text{h, o, u}\}$ denotes the region. For more detailed examples, please refer to Fig.4-(A).

**Region-aware prompts.** As shown in Fig. 3-(B), given a region feature $\mathbf{x}_{(\cdot)} \in \mathbb{R}^d$ and its corresponding concept pool $\mathcal{C}_{(\cdot)}^v$, we compute cosine similarity between the feature and each concept:

$$s_{(\cdot)}^{v,k} = \frac{\langle \mathbf{x}_{(\cdot)}, \mathbf{c}_{(\cdot)}^{v,k} \rangle}{\|\mathbf{x}_{(\cdot)}\| \, \|\mathbf{c}_{(\cdot)}^{v,k}\|}. \tag{11}$$

To highlight the most informative concepts while ignoring irrelevant ones, we apply Sparsemax [50] to the similarity scores, which assigns exact zero weights to uninformative entries and retains only the most relevant concepts. Using the resulting scalar weight $W_{(\cdot)}^{v,k}$, we compute a region concept vector:

$$\bar{\mathbf{c}}_{(\cdot)}^v = \sum_{k=1}^{K} \mathbf{c}_{(\cdot)}^{v,k}\, W_{(\cdot)}^{v,k} \in \mathbb{R}^d. \tag{12}$$

Table 1: Comparison under NF and RF settings. We report harmonic mean (HM) between Unseen and Seen. Best in **bold**, second best underlined.

| Method | Backbone | NF-UC | | | | RF-UC | | | |
|---|---|---|---|---|---|---|---|---|---|
| | | HM | Full | Unseen | Seen | HM | Full | Unseen | Seen |
| GEN-VLKT [23] | Resnet50+ViT-B | 24.17 | 23.71 | 25.05 | 23.38 | 26.08 | 30.56 | 21.36 | 32.91 |
| EoID [52] | Resnet50 | 26.71 | 26.69 | 26.76 | 26.66 | 26.11 | 29.52 | 22.04 | 31.39 |
| HOICLIP [27] | Resnet50+ViT-B | 28.70 | 27.75 | 29.36 | 28.10 | 26.55 | 32.99 | 25.83 | 28.47 |
| ADA-CM [10] | Resnet50+ViT-B | 31.76 | 31.39 | 32.41 | 31.13 | 30.48 | 33.01 | 27.63 | 34.35 |
| CLIP4HOI [26] | Resnet50+ViT-B | 29.54 | 28.90 | 31.44 | 28.26 | 31.23 | **34.08** | 27.88 | **35.48** |
| CMMP [24] | Resnet50+ViT-B | 30.82 | 30.18 | 32.09 | 29.71 | 31.10 | 32.18 | 29.45 | 32.87 |
| EZ-HOI [25] | Resnet50+ViT-B | 31.76 | 31.17 | 33.66 | 30.55 | 31.18 | 33.13 | 29.02 | 34.15 |
| Ours | Resnet50+ViT-B | **33.85** | **32.57** | **36.45** | **31.60** | **32.77** | 33.78 | **31.29** | 34.41 |

This concept vector is then used to augment the diversity-aware prompt $\mathbf{t}^v$, producing the final region-aware prompt as follows:

$$\hat{\mathbf{t}}^v_{(\cdot)} = \mathbf{t}^v + \bar{\mathbf{c}}^v_{(\cdot)}\,\gamma \in \mathbb{R}^d, \tag{13}$$

where $\gamma$ is a scalar controlling the degree of augmentation. The final set of region-aware prompts $\mathbf{T}_{(\cdot)} = [\hat{\mathbf{t}}^1_{(\cdot)}, \ldots, \hat{\mathbf{t}}^V_{(\cdot)}]$ is used to compute classification logits, as described in Eq. (4). This improves discriminability of the model by aligning each prompt with region-specific semantics.

# 4 Experiments

## 4.1 Experimental settings

**Datasets.** We conduct experiments on the HICO-DET benchmark for HOI detection. HICO-DET contains 80 object categories from the COCO dataset [51] and 117 actions, forming 600 HOI classes. It includes 47,776 images, with 38,118 for training and 9,658 for testing.

**Zero-shot setting on HICO-DET.** Following prior works [3, 2, 1, 23], we evaluate under four settings: Non-rare First Unseen Composition (NF-UC), Rare First (RF-UC), Unseen Object (UO), and Unseen Verb (UV). NF-UC and RF-UC define 120 unseen and 480 seen HOI triplets from 600 total, with unseen compositions drawn from head and tail categories, respectively. UO uses 68 object classes to construct 500 seen and 100 unseen triplets. UV withholds 20 out of 117 verb classes, yielding 516 seen and 84 unseen triplets.

**Evaluation metric.** Mean Average Precision (mAP) is used to evaluate the model for HOI detection. Specifically, a sample is regarded as a true positive if two conditions are met: 1) the IoU of both human and object bounding boxes is larger than 0.5, and 2) the HOI triplet prediction is correct.

**Implementation Details.** We follow the standard training setup used in prior zero-shot two-stage HOI detection methods [24, 10, 25], where DETR is first fine-tuned on instance-level annotations from the HICO-DET training split. Unless otherwise noted, we use CLIP ViT-B/16 as the visual backbone. Additional implementation and training details are provided in the supplementary material.

## 4.2 Zero-shot HOI detection

We evaluate our method on four zero-shot HOI detection settings in HICO-DET: NF-UC, RF-UC, UO, and UV. Table. 1, 2, and 3 report mAP (Full / Unseen / Seen), harmonic mean (HM), and trainable parameters (#TP), comparing with state-of-the-art baselines. In NF-UC and RF-UC (Table. 1), our method achieves the best scores across all metrics. In NF-UC, we obtain 36.45 (Unseen) and 33.85 (HM), surpassing CLIP4HOI by **+5.01** (Unseen) and **+4.31** (HM). In RF-UC, we outperform EZ-HOI by **+2.27** (Unseen) and **+1.59** (HM). Our method uses only **4.50M** trainable parameters, much fewer than EZ-HOI (6.85M) and CLIP4HOI (56.7M). In UO (Table. 2), we achieve **36.13** (Unseen) and **34.41** (HM), outperforming CMMP and EZ-HOI by **+1.97** and **+2.27** HM, respectively. In UV (Table. 3), we again achieve the best scores: **26.69** (Unseen), **32.72** (Full), and **29.80** (HM), with a **+1.59** gain on Unseen verbs over EZ-HOI. These results confirm the effectiveness of our *visual diversity-aware prompts* and *region-aware prompts* in the zero-shot HOI detection.

Table 2: Comparison under the UO (Unseen Object) setting. We report harmonic mean (HM) between Unseen and Seen. #TP: trainable parameters. Best in **bold**, second best underlined.

| Method | Setting | Backbone | #TP | HM | Full | Unseen | Seen |
|---|---|---|---|---|---|---|---|
| FCL [1] | UO | Resnet50 | – | 17.65 | 19.87 | 15.54 | 20.74 |
| ATL [2] | UO | Resnet50 | – | 17.79 | 20.47 | 15.11 | 21.54 |
| GEN-VLKT [23] | UO | Resnet50 | 42.05M | 20.11 | 25.63 | 15.01 | 28.92 |
| HOICLIP [27] | UO | Resnet50+ViT-B | 66.18M | 20.32 | 28.53 | 16.30 | 30.99 |
| CLIP4HOI [26] | UO | Resnet50+ViT-B | 56.7M | 31.98 | 32.58 | 31.79 | 32.73 |
| CMMP [24] | UO | Resnet50+ViT-B | 2.30M | 32.44 | 31.59 | 33.76 | 31.15 |
| EZ-HOI [25] | UO | Resnet50+ViT-B | 6.85M | 32.14 | 32.27 | 33.28 | 32.06 |
| Ours | UO | Resnet50+ViT-B | 4.50M | **34.41** | **33.39** | **36.13** | **32.84** |

Table 3: Comparison under the UV (Unseen Verb) setting. We report harmonic mean (HM) between Unseen and Seen. #TP: trainable parameters. Best in **bold**, second best underlined.

| Method | Setting | Backbone | #TP | HM | Full | Unseen | Seen |
|---|---|---|---|---|---|---|---|
| GEN-VLKT [23] | UV | Resnet50+ViT-B | 42.05M | 24.35 | 28.74 | 20.96 | 30.23 |
| EoID [52] | UV | Resnet50 | – | 26.29 | 29.61 | 22.71 | 30.73 |
| HOICLIP [27] | UV | Resnet50+ViT-B | 66.18M | 27.72 | 31.09 | 24.30 | 32.19 |
| CLIP4HOI [26] | UV | Resnet50+ViT-B | 56.7M | 28.35 | 30.42 | 26.02 | 31.14 |
| CMMP [24] | UV | Resnet50+ViT-B | 2.30M | 29.23 | 31.84 | 26.23 | 32.75 |
| EZ-HOI [25] | UV | Resnet50+ViT-B | 6.85M | 29.09 | 32.32 | 25.10 | 33.49 |
| Ours | UV | Resnet50+ViT-B | 4.50M | **29.80** | **32.73** | **26.69** | **33.72** |

Table 4: **Ablation results under four zero-shot settings.** VDP: visual diversity-aware prompts, RAP: region-aware prompts, VDRP: full model with both.

| NF-UC | Full | Unseen | Seen | | RF-UC | Full | Unseen | Seen |
|---|---|---|---|---|---|---|---|---|
| BASE | 28.99 | 31.68 | 28.32 | | BASE | 29.80 | 25.64 | 30.84 |
| + VDP | 30.17 | 32.19 | 29.66 | | + VDP | 31.95 | 29.16 | 32.65 |
| + RAP | 30.43 | 34.93 | 29.30 | | + RAP | 32.43 | 26.46 | 33.93 |
| + VDRP | **32.57** | **36.45** | **31.60** | | + VDRP | **33.78** | **31.29** | **34.41** |

| UO | Full | Unseen | Seen | | UV | Full | Unseen | Seen |
|---|---|---|---|---|---|---|---|---|
| BASE | 28.92 | 28.60 | 30.50 | | BASE | 29.72 | 22.41 | 30.91 |
| + VDP | 31.49 | 33.29 | 31.13 | | + VDP | 31.53 | 23.78 | 32.79 |
| + RAP | 31.85 | 33.90 | 31.44 | | + RAP | 31.28 | 24.53 | 32.38 |
| + VDRP | **33.39** | **36.13** | **32.84** | | + VDRP | **32.73** | **26.69** | **33.72** |

## 4.3 Ablation studies

**Component analysis.** To assess each component's contribution, we conduct an ablation study across four zero-shot HOI settings (Table 4). Starting from a static prompt baseline, we add two modules: (1) **VDP** models intra-class variation using group-wise visual variance and Gaussian noise, and (2) **RAP** augments prompt embeddings with retrieved region-specific concepts. **VDP** improves generalization by capturing the variance of diverse verb appearances, while **RAP** enhances discrimination among visually similar verbs. The full model **VDRP** achieves the best performance across all settings, demonstrating the complementary benefits of both modules.

**Group-wise modeling in VDP.** To evaluate the impact of group-wise visual variance, we compare different configurations in the **VDP** module based on the number of verbs per group: (1) a single global variance shared across all verbs (`Global`), (2) group-wise variance using clusters of 3 verbs each (`Group_3`), and (3) clusters of 5 verbs each (`Group_5`). As shown in Table 5, we observe that group-wise variance modeling generally improves over a global estimate, but performance varies depending on the group size.

**Impact of injection scale.** We evaluate variance injection scale $\alpha \in \{0.01, 0.02, 0.10\}$ to analyze its influence on the delta scaling in Eq. 8. Results across all settings are summarized in Table 6. We

Table 5: Impact of group configuration in visual diversity-aware prompts (VDP). Best results in **bold**.

| Method | NF-UC | | | RF-UC | | | UO | | | UV | | |
|---|---|---|---|---|---|---|---|---|---|---|---|---|
| | Full | Unseen | Seen | Full | Unseen | Seen | Full | Unseen | Seen | Full | Unseen | Seen |
| Global | 31.92 | 35.92 | 30.92 | **33.94** | **32.27** | 34.36 | 32.30 | 34.23 | 31.91 | 32.55 | 26.07 | 33.60 |
| Group_3 | 32.43 | 35.89 | 31.57 | 33.15 | 29.57 | 34.04 | 32.70 | 34.76 | 32.29 | 32.75 | 26.25 | **33.81** |
| Group_5 | **32.57** | **36.45** | **31.60** | 33.78 | 31.29 | **34.41** | **33.39** | **36.13** | **32.84** | **32.73** | **26.69** | 33.72 |

Table 6: Impact of injection scale $\alpha$ on performance across zero-shot settings. Best results in **bold**.

| $\alpha$ | NF-UC | | | RF-UC | | | UO | | | UV | | |
|---|---|---|---|---|---|---|---|---|---|---|---|---|
| | Full | Unseen | Seen | Full | Unseen | Seen | Full | Unseen | Seen | Full | Unseen | Seen |
| 0.01 | 32.03 | 35.29 | 31.22 | 33.51 | 30.58 | 34.24 | 32.81 | 35.52 | 32.27 | 32.48 | 25.97 | 33.54 |
| 0.10 | 32.02 | 35.80 | 31.07 | 33.16 | 29.86 | 33.98 | 32.81 | 35.11 | 32.35 | 32.54 | 24.73 | **33.81** |
| 0.02 | **32.57** | **36.45** | **31.60** | **33.78** | **31.29** | **34.41** | **33.39** | **36.13** | **32.84** | **32.73** | **26.69** | 33.72 |

Table 7: Effect of Gaussian perturbation across four zero-shot settings. Best results are in **bold**.

| Method | NF-UC | | | | RF-UC | | | |
|---|---|---|---|---|---|---|---|---|
| | HM | Full | Unseen | Seen | HM | Full | Unseen | Seen |
| W/O perturbation | 33.57 | 32.49 | 35.68 | **31.69** | 31.76 | 33.12 | 29.83 | 33.95 |
| W/ perturbation | **33.85** | **32.57** | **36.45** | 31.60 | **32.77** | **33.78** | **31.29** | **34.41** |

| Method | UO | | | | UV | | | |
|---|---|---|---|---|---|---|---|---|
| | HM | Full | Unseen | Seen | HM | Full | Unseen | Seen |
| W/O perturbation | 33.67 | 33.09 | 34.59 | 32.79 | 29.49 | 32.72 | 26.16 | **33.79** |
| W/ perturbation | **34.41** | **33.39** | **36.13** | **32.84** | **29.80** | **32.73** | **26.69** | 33.72 |

observe that $\alpha = 0.02$ consistently achieves the best trade-off between unseen and seen performance. This value was chosen to match the initialization scale of CLIP's context embeddings, ensuring stability as excessively large $\alpha$ causes over-perturbed context tokens and unstable training. Given that context tokens are highly sensitive to initialization, keeping the scale aligned promotes stable optimization while maintaining sufficient diversity.

**Effect of Gaussian perturbation.** We study the effect of Gaussian perturbation on prompt embeddings by varying the scale $\beta \in \{0, 0.1\}$. Table 7 summarizes the results across all zero-shot settings. Perturbation improves generalization in most unseen cases, confirming that stochastic sampling helps the model capture diverse visual realizations of each verb.

**Retrieval strategy.** We compare three retrieval methods for region concepts: Softmax, Top-3, and Sparsemax (Table 8). Overall, Sparsemax performs most consistently, benefiting from its ability to suppress irrelevant concepts and emphasize informative ones.

**Impact of augmentation scaling factor.** We evaluate $\gamma \in \{0.2, 0.5, 1.0\}$ in Eq. 13 to study how strongly region-derived cues contribute to prompt adaptation. Results are reported in Table 9. A moderate scaling of $\gamma = 0.2$ provides the best overall balance, whereas excessive cue weighting ($\gamma = 1.0$) slightly degrades performance across multiple settings, likely due to noisy region concepts.

## 4.4 Qualitative results

Fig. 4 shows qualitative examples of region-specific concept generation and retrieval. Given a verb prompt (e.g., "licking an object"), our method uses LLMs (e.g., LLaMA-7B [48] and GPT-4 [49]) to generate $K$ region concepts, capturing fine-grained concepts such as facial expression or visual patterns. While LLaMA-7B provides diverse concept descriptions, we observed noise and redundancy in object-related concepts; therefore, GPT-4 was additionally used (see Supplementary for details). As illustrated, similar verbs like "licking" and "eating" are associated with distinct region-specific signals—enabling the model to better distinguish subtle visual differences, enhancing the discriminability.

Table 8: Comparison of retrieval strategies for region-aware prompts (RAP). Best results in **bold**.

| Method | NF-UC | | | RF-UC | | | UO | | | UV | | |
|---|---|---|---|---|---|---|---|---|---|---|---|---|
| | Full | Unseen | Seen | Full | Unseen | Seen | Full | Unseen | Seen | Full | Unseen | Seen |
| Softmax($\cdot$) | 32.01 | 35.93 | 31.03 | 33.32 | 30.37 | 34.06 | 32.91 | 35.83 | 32.32 | 32.57 | 25.45 | **33.74** |
| Top-3 | 31.97 | 35.85 | 31.00 | 33.35 | 30.80 | 33.99 | 32.94 | 33.57 | 32.41 | 32.47 | 25.85 | 33.55 |
| Sparsemax($\cdot$) | **32.57** | **36.45** | **31.60** | **33.78** | **31.29** | **34.41** | **33.39** | **36.13** | **32.84** | **32.73** | **26.69** | 33.72 |

Table 9: Effect of augmentation scale $\gamma$ on performance across all settings. Best results in **bold**.

| $\gamma$ | NF-UC | | | RF-UC | | | UO | | | UV | | |
|---|---|---|---|---|---|---|---|---|---|---|---|---|
| | Full | Unseen | Seen | Full | Unseen | Seen | Full | Unseen | Seen | Full | Unseen | Seen |
| 0.5 | 32.14 | 35.92 | 31.19 | 33.52 | 30.73 | 34.21 | 33.08 | 35.17 | 32.66 | 32.72 | **27.06** | 33.64 |
| 1.0 | 31.85 | 35.15 | 31.02 | 33.48 | 30.67 | 34.18 | 32.60 | 34.74 | 32.17 | 32.54 | 25.92 | 33.62 |
| 0.2 | **32.57** | **36.45** | **31.60** | **33.78** | **31.29** | **34.41** | **33.39** | **36.13** | **32.84** | **32.73** | 26.69 | **33.72** |

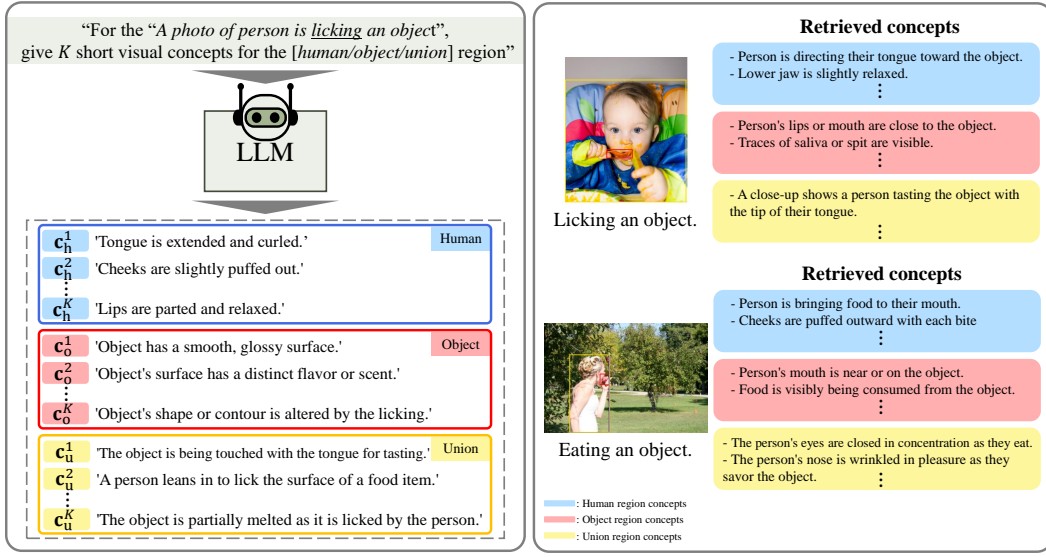

Figure 4: **Qualitative examples of region concept generation and retrieval.** (A) Given a verb prompt and region type, an LLM generates $K$ region concepts per verb. (B) Retrieved concepts for "Licking" and "Eating" highlight subtle region concepts that help disambiguate visually similar interactions. Concepts are color-coded by region: blue (human), red (object), yellow (union).

## 5 Conclusion

We present **VDRP**, a prompt learning framework for zero-shot HOI detection that addresses the *visual complexity of interaction*, including intra-class diversity and inter-class entanglement among verb classes. To this end, we propose a *visual diversity-aware prompt learning* that injects group-wise visual variance into the prompt context and applies Gaussian perturbation, enabling prompt embeddings to better reflect diverse visual appearances. We also introduce *region-aware prompt augmentation*, which augments diversity-aware prompts with region-specific concepts from the human, object, and union regions to improve verb discriminability. Extensive experiments on HICO-DET show that our method outperforms prior work across four zero-shot settings while maintaining high parameter efficiency. These results underscore the value of combining distributional modeling and region-level augmentation for generalizable zero-shot HOI detection.

# 6 Acknowledgments

This work was partly supported by Korea Research Institute for defense Technology planning and advancement - Grant funded by Defense Acquisition Program Administration (DAPA) (KRIT-CT-23-021, 45%), by the Institute of Information & Communications Technology Planning & Evaluation (IITP)-ITRC (Information Technology Research Center) grant funded by the Korea government (MSIT) (IITP-2025-RS-2024-00436857, 45%) and by the National Research Foundation of Korea (RS-2022-NR068758, 10%). We thank **Jaewoon Byun** for helpful discussion.

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

# Appendix

## A  Implementation details

We conduct all experiments using PyTorch with mixed-precision training. For the base model using CLIP ViT-B/16, we train on two NVIDIA GeForce RTX 3090 GPUs with a batch size of 8 for 12 epochs. We use the AdamW [53] optimizer with an initial learning rate of $1 \times 10^{-3}$, decayed to $1 \times 10^{-4}$ using a cosine scheduler and a weight decay of 8. We report the best results after applying weight decay. The framework adopts CLIP ViT-B/16 as the visual encoder, where the adapter applies a bottleneck transformation from $d_{\text{up}} = 768$ to $d_{\text{down}} = 64$, followed by projection into a shared embedding space of dimension $d = 512$. Human-object detection is performed by a frozen DETR model with Resnet50 [54], and detections with confidence below the threshold $\theta = 0.2$ are discarded. The context embedding consists of $N_{\text{ctx}} = 24$ learnable tokens, each initialized from a Gaussian distribution with standard deviation 0.02. To incorporate group-wise variance, we inject a modulation vector (scaled by $\alpha = 0.02$) into the context embedding and apply Gaussian perturbation to the resulting prompt embedding with a noise scale $\beta = 0.1$. For region-aware prompt augmentation, we generate $K = 10$ region concepts per verb and region type (human, object, union) using LLaMA-7B [48] and ChatGPT-4 [49]. These are encoded by the frozen CLIP text encoder and aggregated via Sparsemax [50] to form a concept vector, which is added to the prompt embedding with scaling factor $\gamma = 0.2$. We additionally conduct experiments with a scaled-up version using CLIP ViT-L/14 as the visual encoder. In this setting, the adapter transforms $d_{\text{up}} = 1024$ to $d_{\text{down}} = 64$, with the final embedding dimension set to $d = 768$. Due to memory constraints, these experiments are performed on two NVIDIA RTX 6000 Ada Generation GPUs with a reduced batch size of 4 per GPU.

## B  Details of our architectures

**Details of spatial head.** To enhance the union-region representation with geometric priors, we design a spatial head that encodes the spatial relationship between the human and object bounding boxes and fuses it with the region features, inspired by prior works [55, 5, 6]. Given a human box $b_{\text{h}} = [x_1, y_1, x_2, y_2]$ and an object box $b_{\text{o}} = [x_1', y_1', x_2', y_2']$, we compute the center coordinates, widths, and heights:

$$\mathbf{C}_{\text{h}} = \left( \frac{x_1 + x_2}{2}, \frac{y_1 + y_2}{2} \right), \quad \mathbf{C}_{\text{o}} = \left( \frac{x_1' + x_2'}{2}, \frac{y_1' + y_2'}{2} \right),$$

$$w_{\text{h}} = x_2 - x_1, \quad h_{\text{h}} = y_2 - y_1, \quad w_{\text{o}} = x_2' - x_1', \quad h_{\text{o}} = y_2' - y_1'.$$

We then extract spatial features including normalized positions, relative box areas, aspect ratios, intersection-over-union (IoU), and direction-aware relative distances:

$$d_x = \frac{|\mathbf{C}_{\text{o}x} - \mathbf{C}_{\text{h}x}|}{w_{\text{h}} + \epsilon}, \quad d_y = \frac{|\mathbf{C}_{\text{o}y} - \mathbf{C}_{\text{h}y}|}{h_{\text{h}} + \epsilon}.$$

These values are concatenated and passed through a small feedforward network to produce a spatial encoding vector $\mathbf{E}_S$. To incorporate this geometric context, we apply a multi-modal fusion module $\phi_{\text{MMF}}(\cdot)$ that combines the human and object features $[\mathbf{x}_{\text{h}}; \mathbf{X}_{\text{o}}]$ with the spatial encoding:

$$\mathbf{f}_S = \phi_{\text{MMF}} \left( [\mathbf{x}_{\text{h}}; \mathbf{x}_{\text{o}}], \ \mathbf{E}_S \right).$$

Finally, the spatial query $\mathbf{f}_S$ is concatenated with the union feature $\mathbf{x}_{\bar{\text{u}}}$, and passed through a projection layer to obtain the final spatially-enhanced representation:

$$\mathbf{x}_{\text{u}} = \text{Proj} \left( [\mathbf{x}_{\bar{\text{u}}}; \mathbf{f}_S] \right),$$

which is used for verb classification with visual diversity and region-aware prompts.

**Details of sparsemax.** To select region-level concepts relevant to each human-object interaction, we adopt the Sparsemax [50] activation function in our region-aware prompt augmentation module. Unlike Softmax, which produces dense probability distributions where all entries are non-zero, Sparsemax enables *sparse* selection by assigning exact zeros to irrelevant entries. This property is particularly useful in our setting, as only a subset of the $K$ generated region concepts are semantically aligned with the visual feature.

Formally, given a score vector $\mathbf{s} \in \mathbb{R}^K$—representing the cosine similarity between a region feature and $K$ region concept embeddings—we define Sparsemax as a projection onto the probability simplex:

$$\text{Sparsemax}(\mathbf{s}) := \arg\min_{\mathbf{a} \in \Delta^K} \|\mathbf{a} - \mathbf{s}\|^2, \quad \Delta^K = \left\{ \mathbf{a} \in \mathbb{R}^K \mid \sum_{k=1}^{K} a_k = 1, \ a_k \geq 0 \right\}. \quad (14)$$

This formulation yields a closed-form solution that projects the input vector onto the probability simplex $\Delta^K$, resulting in a sparse vector $\mathbf{a}$ where low-scoring entries are assigned exact zeros. We apply Sparsemax to the similarity scores between each region feature and the corresponding concept pool, allowing the model to focus on the most informative region concepts while ignoring noisy or irrelevant ones.

## C   Limitations

While our method demonstrates strong performance across multiple zero-shot HOI detection settings, several broader limitations remain. First, the *region-aware prompts* (RAP) module builds upon region-level concepts generated by large language models (LLMs). Although effective in capturing contextual semantics, LLM-based concepts may be noisy or misaligned with visual concepts due to inherent limitations in language–vision grounding. Improving robustness through confidence-aware filtering or vision-aligned concept refinement is a promising direction. Second, our framework builds on prompt learning, which presents structural challenges when applied to a large number of classes or diverse interaction types. Prompt representations can be sensitive to scaling strategies, initialization, and optimization dynamics, making them less stable in settings with limited data or rare class compositions. This reflects a broader limitation of prompt-based models, where generalization can be affected by prompt granularity, representation collapse, or lack of compositional structure. Future directions may include more robust prompt initialization, adaptive scaling mechanisms, or compositional prompt construction to improve stability and generalization.

## D   Broader impacts

Our work builds upon vision-language models (VLMs) such as CLIP, which are trained on large-scale web-scraped image-text pairs. As a result, our model may inadvertently inherit biases present in the pretraining data, such as cultural stereotypes or imbalanced representations across demographic groups. While our method aims to improve generalization in zero-shot HOI detection, deployment in real-world applications should be done with caution—particularly in contexts involving surveillance, behavioral analysis, or human activity interpretation. Furthermore, the retrieval-based region prompt augmentation using large language models (e.g., LLaMA-7B and ChatGPT-4), which may also reflect biases or hallucinated associations. Misinterpretation of human-object interactions in sensitive domains (e.g., law enforcement or healthcare) could lead to harmful outcomes if such biases are not properly addressed. To mitigate potential misuse, we recommend restricting deployment to controlled environments with human oversight, and further advocate for evaluating fairness metrics across subpopulations when applying the model to downstream applications.

## E   Further experiments

**V-COCO datasets.** V-COCO [56] dataset has 80 object categories derived from COCO datasets [51] and includes 29 interactions, resulting 263 HOI compositions. The number of image samples is 10,396 (5,400/4,964 for train/test).

**Fully supervised settings.**    We further evaluate our method under the fully supervised HOI detection setting on HICO-DET and V-COCO, as summarized in Table 10. Among two-stage methods, our model achieves the best performance on HICO-DET, with an mAP of 39.07, surpassing the prior state-of-the-art EZ-HOI [25]. While one-stage methods such as UniHOI and AGER benefit from end-to-end optimization and full supervision of all components, they typically rely on larger networks and longer training schedules. In contrast, our two-stage approach uses fewer learnable parameters and is designed primarily for zero-shot generalization, which likely limits the effectiveness of group-wise variance modeling and hinders the learning of cross-class structure. On

Table 10: State-of-the-art comparison on HICO-DET and V-COCO under the fully-supervised setting. Bold indicates the best-performing method within each group (one-stage vs. two-stage).

| Method | HICO-DET | | | V-COCO | |
|---|---|---|---|---|---|
| | Full | Rare | Nonrare | $AP^{s1}_{role}$ | $AP^{s2}_{role}$ |
| *One-stage Methods* | | | | | |
| GEN-VLKT [23] | 33.75 | 29.25 | 35.10 | 62.4 | 64.5 |
| HOICLIP [27] | 34.69 | 31.12 | 35.74 | 63.5 | 65.0 |
| RLIPV2 [57] | 35.38 | 29.61 | 37.11 | **65.9** | 68.0 |
| AGER [58] | 36.75 | 33.53 | 37.43 | 65.7 | 67.9 |
| LogicHOI [59] | 35.47 | 32.03 | 36.64 | 64.6 | 65.6 |
| UniHOI [11] | **40.06** | **39.91** | **40.11** | 65.6 | **68.3** |
| *Two-stage Methods* | | | | | |
| UPT [5] | 32.62 | 28.62 | 33.81 | 59.0 | 64.5 |
| ADA-CM [10] | 38.40 | 37.52 | 38.66 | 58.6 | 64.0 |
| CLIP4HOI [26] | 35.33 | 33.95 | 35.75 | – | **66.3** |
| CMMP [24] | 38.14 | 37.75 | 38.25 | – | 64.0 |
| EZ-HOI [25] | 38.61 | 37.70 | 38.89 | 60.5 | 66.2 |
| **Ours** | **39.07** | **39.08** | **39.06** | **60.6** | 66.2 |

Table 11: Zero-shot HOI detection results under four splits—NF-UC, RF-UC, UO, and UV—using the scaled CLIP (ViT-L). HM is harmonic mean (HM) between Unseen and Seen. Best results are shown in **bold**, and second best are underlined.

| Method | NF-UC | | | | RF-UC | | | |
|---|---|---|---|---|---|---|---|---|
| | HM | Full | Unseen | Seen | HM | Full | Unseen | Seen |
| UniHOI [11] | 30.40 | 31.79 | 28.45 | 32.63 | 30.76 | 32.27 | 28.68 | 33.16 |
| CMMP [24] | 34.50 | 35.13 | 33.52 | 35.53 | 36.69 | 37.13 | 35.98 | 37.42 |
| EZ-HOI [25] | 35.38 | 34.84 | 36.33 | 34.47 | 35.73 | 36.73 | 34.24 | 37.35 |
| **Ours** | **36.83** | **36.46** | **37.48** | **36.21** | **37.58** | **38.13** | **36.72** | **38.48** |

| Method | UO | | | | UV | | | |
|---|---|---|---|---|---|---|---|---|
| | HM | Full | Unseen | Seen | HM | Full | Unseen | Seen |
| UniHOI [11] | 25.17 | 31.56 | 19.72 | 34.76 | 30.50 | 34.68 | 26.05 | 36.78 |
| CMMP [24] | 37.83 | 36.74 | **39.67** | 36.15 | 33.75 | 36.38 | 30.84 | 37.28 |
| EZ-HOI [25] | 37.06 | 36.38 | 38.17 | 36.02 | 32.84 | 36.84 | 28.82 | 38.15 |
| **Ours** | **38.41** | **37.81** | 39.36 | **37.50** | **34.31** | **37.18** | **31.16** | **38.16** |

the V-COCO benchmark, our model achieves 60.6 and 66.2 AP under Scenario 1 and Scenario 2, respectively—comparable to other two-stage models. We attribute this to the small number of verb classes (24 vs. 117 in HICO-DET) and limited dataset size, which reduce the effectiveness of group-wise variance modeling and cross-class structure learning. Overall, these results demonstrate that although our method is designed to address zero-shot HOI detection, it generalizes well to fully supervised settings, retaining strong performance even when supervision is abundant.

**Scaled-up CLIP (ViT-L) setting.** To assess the effect of scaling the vision backbone, we evaluate our method using the CLIP ViT-L/14 encoder in place of the default ViT-B/16. As shown in Table 11, our model consistently improves performance across all four zero-shot evaluation splits—NF-UC, RF-UC, UO, and UV. Notably, our method achieves the highest harmonic mean (HM) across all settings, reflecting balanced generalization to both seen and unseen verb compositions. While prior methods such as CMMP [24] and EZ-HOI [25] exhibit competitive results on specific splits, their performance fluctuates across evaluation scenarios. In contrast, our scaled-up model maintains stable gains throughout, demonstrating that visual diversity-aware prompt learning and region-aware prompt augmentation remain effective even when paired with high-capacity vision-language encoders. These results highlight the scalability of our framework even when scaled to stronger backbones.

Table 12: Comparison of using both mean and variance $(\mu, \sigma^2)$ vs. variance-only $(\sigma^2)$ in the VDP across four zero-shot settings. Best in **bold**.

| Setting | Stats | Full | Unseen | Seen |
|---|---|---|---|---|
| NF-UC | $(\mu, \sigma^2)$ | 32.03 | 35.75 | 31.10 |
| | $\sigma^2$ only | **32.57** | **36.45** | **31.60** |
| RF-UC | $(\mu, \sigma^2)$ | 33.19 | 30.34 | 33.91 |
| | $\sigma^2$ only | **33.78** | **31.29** | **34.41** |
| UO | $(\mu, \sigma^2)$ | 33.03 | 34.87 | 32.66 |
| | $\sigma^2$ only | **33.39** | **36.13** | **32.84** |
| UV | $(\mu, \sigma^2)$ | 32.59 | 25.33 | **33.77** |
| | $\sigma^2$ only | **32.73** | **26.69** | 33.72 |

Table 13: Ablation study comparing region branches (Human, Object and Union) and the full model (H+O+U) under different zero-shot settings. Best in **bold**.

| Branch | NF-UC | | | | RF-UC | | | |
|---|---|---|---|---|---|---|---|---|
| | HM | Full | Unseen | Seen | HM | Full | Unseen | Seen |
| Human | 33.27 | 32.03 | 35.77 | 31.09 | 32.19 | 33.34 | 30.53 | 34.04 |
| Object | 33.48 | 32.17 | 36.16 | 31.17 | 31.90 | 33.32 | 29.92 | 34.17 |
| Union | 33.26 | 32.17 | 35.40 | 31.37 | 32.01 | 33.23 | 30.28 | 33.96 |
| H+O+U | **33.85** | **32.57** | **36.45** | **31.60** | **32.77** | **33.78** | **31.29** | **34.41** |

| Branch | UO | | | | UV | | | |
|---|---|---|---|---|---|---|---|---|
| | HM | Full | Unseen | Seen | HM | Full | Unseen | Seen |
| Human | 34.08 | 32.90 | 36.11 | 32.26 | 29.35 | 32.66 | 25.97 | 33.75 |
| Object | 33.39 | 32.65 | 34.61 | 32.26 | 29.64 | **33.07** | 26.15 | **34.20** |
| Union | 33.39 | 32.70 | 34.52 | 32.34 | 29.55 | 32.60 | 26.36 | 33.62 |
| H+O+U | **34.41** | **33.39** | **36.13** | **32.84** | **29.80** | 32.73 | **26.69** | 33.72 |

# F  More ablation studies

**Effect of mean vs. variance in VDP.** To evaluate the contribution of distributional statistics in our visual diversity-aware prompts (VDP) module, we compare two variants: one that uses only group-wise variance ($\sigma^2$), and another that combines both the group-wise mean ($\mu$) and variance ($\sigma^2$) through concatenation followed by projection. Table 12 reports results across all four zero-shot settings. Overall, we find that the variance-only variant consistently matches or outperforms the mean-variance combination—especially on the Unseen splits in all settings. For instance, in NF-UC, using variance alone yields the highest unseen mAP of **36.45**, compared to 35.75 with $\mu+\sigma^2$. This suggests that variance alone serves as a stronger signal for modeling intra-class visual diversity, particularly under zero-shot generalization conditions. Prior studies such as Zhu et al. [40] also report that variance-centered prompt representations improve generalizability by capturing the dispersion of visual features without overfitting to sample means. Our results align with this observation, indicating that the inclusion of the mean may introduce redundant or unstable signals—especially for low-shot or noisy verb clusters, where the class prototype is poorly defined. An exception arises in the UV setting, where the combined variant slightly improves performance on seen metrics. Nonetheless, the overall trend supports the effectiveness and robustness of variance-only modeling for capturing distributional diversity in the prompt space.

**Effect of branches in RAP.** We compare the individual contributions of the human, object, and union region branches under four zero-shot HOI detection settings. The results are summarized in Table 13. Across all settings, the full model consistently outperforms each individual branch, confirming the complementary nature of region-level concepts. Notably, the proposed combination achieves the best unseen mAP in all settings, demonstrating the effectiveness of region-aware prompts in enhancing discriminability. These findings highlight that human and object concepts capture distinct but complementary contextual signals, while the union branch provides holistic spatial grounding that reinforces interaction understanding.

Table 14: Zero-shot verb variance robustness under the UV setting. We remove the top-3 semantic neighbors for each unseen verb before computing visual variance.

|          | HM    | Full  | Unseen | Seen  |
|----------|-------|-------|--------|-------|
| W/O Top-3 | 29.48 | 32.64 | 26.21 | 33.69 |
| W/ Top-3  | **29.80** | **32.73** | **26.69** | **33.72** |

Table 15: Sensitivity analysis of $\tau$-Sparsemax under four zero-shot settings. Best results are in **bold**.

| $\tau$ value | **NF-UC** | | | **RF-UC** | | |
|---|------|--------|------|------|--------|------|
|     | Full  | Unseen | Seen  | Full  | Unseen | Seen  |
| 0.10 | 32.23 | 35.85 | 31.33 | 33.59 | 30.92 | 34.26 |
| 0.05 | 32.38 | 35.76 | 31.53 | 33.10 | 29.56 | 33.99 |
| 0.00 | **32.57** | **36.45** | **31.60** | **33.78** | **31.29** | **34.41** |

| $\tau$ value | **UO** | | | **UV** | | |
|---|------|--------|------|------|--------|------|
|     | Full  | Unseen | Seen  | Full  | Unseen | Seen  |
| 0.10 | 32.90 | 35.84 | 32.31 | **32.85** | **26.95** | **33.81** |
| 0.05 | 32.95 | 34.63 | 32.62 | 32.73 | 25.91 | 33.84 |
| 0.00 | **33.39** | **36.13** | **32.84** | 32.73 | 26.69 | 33.72 |

Table 16: Robustness of VDP under few-shot variance sampling ($N_v = 5$). Best results are in **bold**.

| **NF-UC** | Full  | Unseen | Seen  | **RF-UC** | Full  | Unseen | Seen  |
|-----------|-------|--------|-------|-----------|-------|--------|-------|
| Few-shot  | 32.23 | 35.90 | 31.31 | Few-shot  | 33.40 | 30.59 | 34.10 |
| All       | **32.57** | **36.45** | **31.60** | All       | **33.78** | **31.29** | **34.41** |

| **UO** | Full  | Unseen | Seen  | **UV** | Full  | Unseen | Seen  |
|--------|-------|--------|-------|--------|-------|--------|-------|
| Few-shot | 32.55 | 34.75 | 32.11 | Few-shot | 32.89 | 25.88 | **34.04** |
| All      | **33.39** | **36.13** | **32.84** | All      | **32.73** | **26.69** | 33.72 |

# G   Robustness of visual diversity-aware prompt learning

**Zero-shot verb variance robustness.**   To evaluate the robustness of group-wise variance estimation for unseen verbs, we conduct an additional experiment under the UV setting. Specifically, we remove the top-3 semantic neighbors (based on CLIP similarity) for each unseen verb from the training set before computing the group-wise visual variance. This setup simulates a scenario where certain verbs lack semantically related support, allowing us to analyze the stability of variance estimation under reduced contextual guidance. As shown in Tab 14, the removal of neighboring classes results in only a marginal performance drop in unseen verbs (-0.48 mAP). This indicates that the proposed grouping mechanism maintains its generalization ability even when semantic support is limited.

**$\tau$-Sparsemax sensitivity.**   To investigate the effect of sparsity calibration in concept retrieval, we introduce $\tau$-Sparsemax($\cdot$), where all values below a threshold $\tau \in \{0.0, 0.05, 0.1\}$ are zeroed post-Sparsemax($\cdot$). This modification controls the degree of sparsity in the region-concept selection process. The results in Tab 15 show that excessive pruning ($\tau = 0.05$) slightly decreases overall performance, while higher sparsity ($\tau = 0.10$) recovers seen-class precision at the cost of unseen generalization. The default $\tau = 0.0$ provides the most balanced outcome, indicating that the original Sparsemax($\cdot$) formulation already achieves an effective level of adaptive sparsity.

**Robustness of visual variance under few-shot sampling.**   We further test the stability of visual variance estimation when fewer visual samples are available. For this experiment, group-wise covariance matrices are computed using only five randomly sampled features per verb ($N_v = 5$), compared to using all available samples. As summarized in Tab 16, the performance reduction across all settings is minimal (typically less than 0.6 mAP), demonstrating that VDP remains robust even under limited sample diversity.

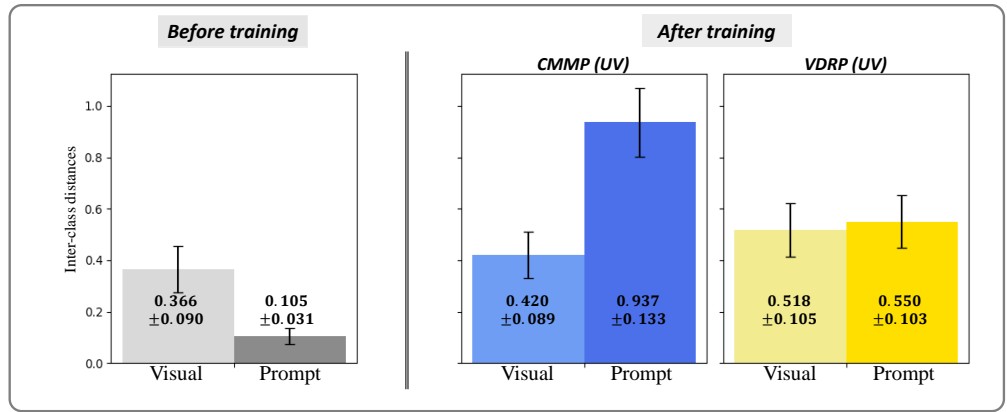

Figure 5: **Pairwise inter-class distances between prompts and visual features.** We report the average pairwise cosine distance (i.e., $D = \mathbb{E}_{i \neq j}[1 - \cos(z_i, z_j)]$) across verb classes, for both visual and prompt embeddings, before and after training. Visual features are extracted from union regions. Before training, we use the CLS token from the CLIP visual encoder applied to cropped union region images. After training, we follow the RoI-Align feature extraction pipeline consistent with the two-stage method (i.e., pooling patch embeddings within the union box). For each verb class, a medoid is selected among all union features to represent its prototype. While prompt embeddings are initially collapsed with low diversity, VDRP maintains a balanced and aligned distribution relative to visual features, unlike CMMP which over-separates prompts and disrupts cross-modal structure.

## H  Inter-class alignment of visual and prompt representations

To assess inter-class alignment between visual and prompt representations, we analyze the distributional structure of each modality after training. Specifically, we examine whether our visual diversity-aware prompt learning preserves class-level relationships that are consistent with those observed in the visual embedding space. We measure the pairwise cosine distance between class-level prototypes in both visual and prompt embedding spaces:

$$D = \mathbb{E}_{i \neq j}[1 - \cos(z_i, z_j)],$$

where $z_i$ denotes the prototype embedding of the $i$-th verb class. For the visual side, we extract features from the union region of each verb instance using the CLIP visual encoder. Before training, we use the CLS token of the encoder applied directly to the union-cropped image. After training, we adopt a region-specific pooling approach: patch embeddings within the union RoI-Align [44] are aggregated to match the training architecture of our method. To mitigate over-smoothing effects from averaging, we define the visual prototype for each class as the *medoid*—the instance whose embedding minimizes the average distance to others within the same class. For the prompt side, we collect the final trained verb embeddings for each class. Fig. 5 presents the average inter-class distance for both modalities across three conditions: (1) before training, (2) after training with CMMP [24], and (3) after training with our method (VDRP). We observe that CMMP significantly increases prompt diversity (0.937) while visual diversity remains moderate (0.420), indicating a modality mismatch. In contrast, VDRP maintains a comparable level of diversity in both modalities (prompt: 0.550, visual: 0.518), suggesting improved distributional alignment. This outcome implies that our prompt learning strategy—though primarily guided by intra-class visual variance—can indirectly induce a structured inter-class layout without over-amplifying semantic separation. We draw inspiration from recent works [60, 43, 36, 39, 40, 37], which emphasize distributional approaches to improving visual–prompt alignment. Motivated by this perspective, we interpret the improved visual–prompt alignment as a natural byproduct of our variance-aware design, even though cross-modal matching was not explicitly enforced.

**Discussion.** This inter-class alignment result further substantiates the novelty of our visual diversity-aware design. Unlike prior distribution-based prompt learning methods such as ProDA [37] and DAPT [36], which regularize textual embeddings, VDP explicitly grounds the prompt distribution in the visual modality by injecting group-wise variance extracted from union-region features. This design enables structured yet balanced prompt distributions that mirror the visual embedding space, achieving semantic coherence without explicit regularization losses. Consequently, VDP naturally promotes inter-class consistency and zero-shot generalization—key objectives that motivated our dual-module framework.

# I Qualitative Results

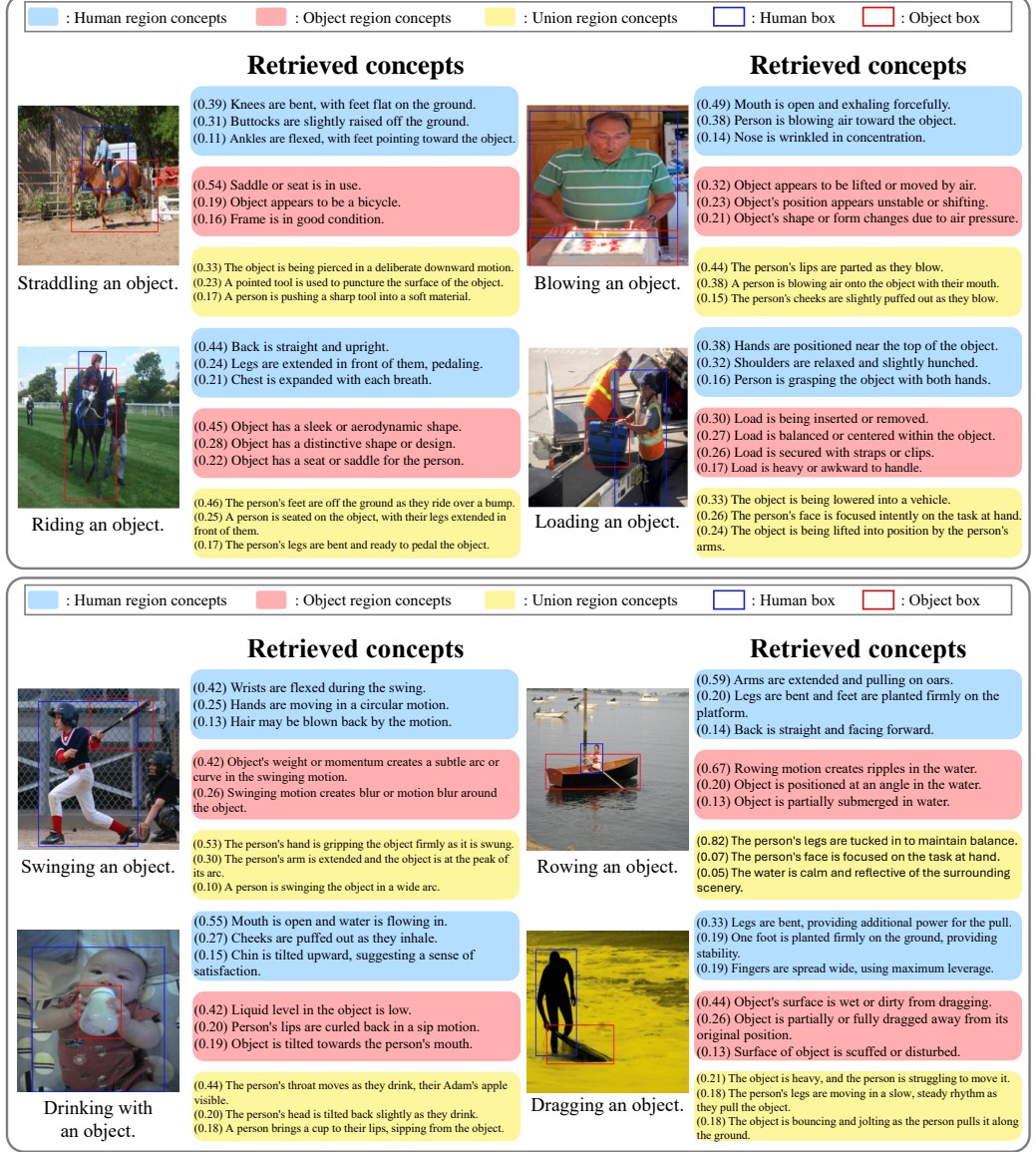

Figure 6: **Region-aware concept retrieval results (1/2).** Each row shows retrieved human, object, and union concepts. Concepts are color-coded by region: blue (human), red (object), yellow (union).

We present qualitative examples in Fig. 6 and 7 to illustrate how region-specific concepts contribute to verb classification. Each example visualizes the top-weighted concepts retrieved from three region branches—human, object, and union—based on sparsemax-normalized concept weights. The full

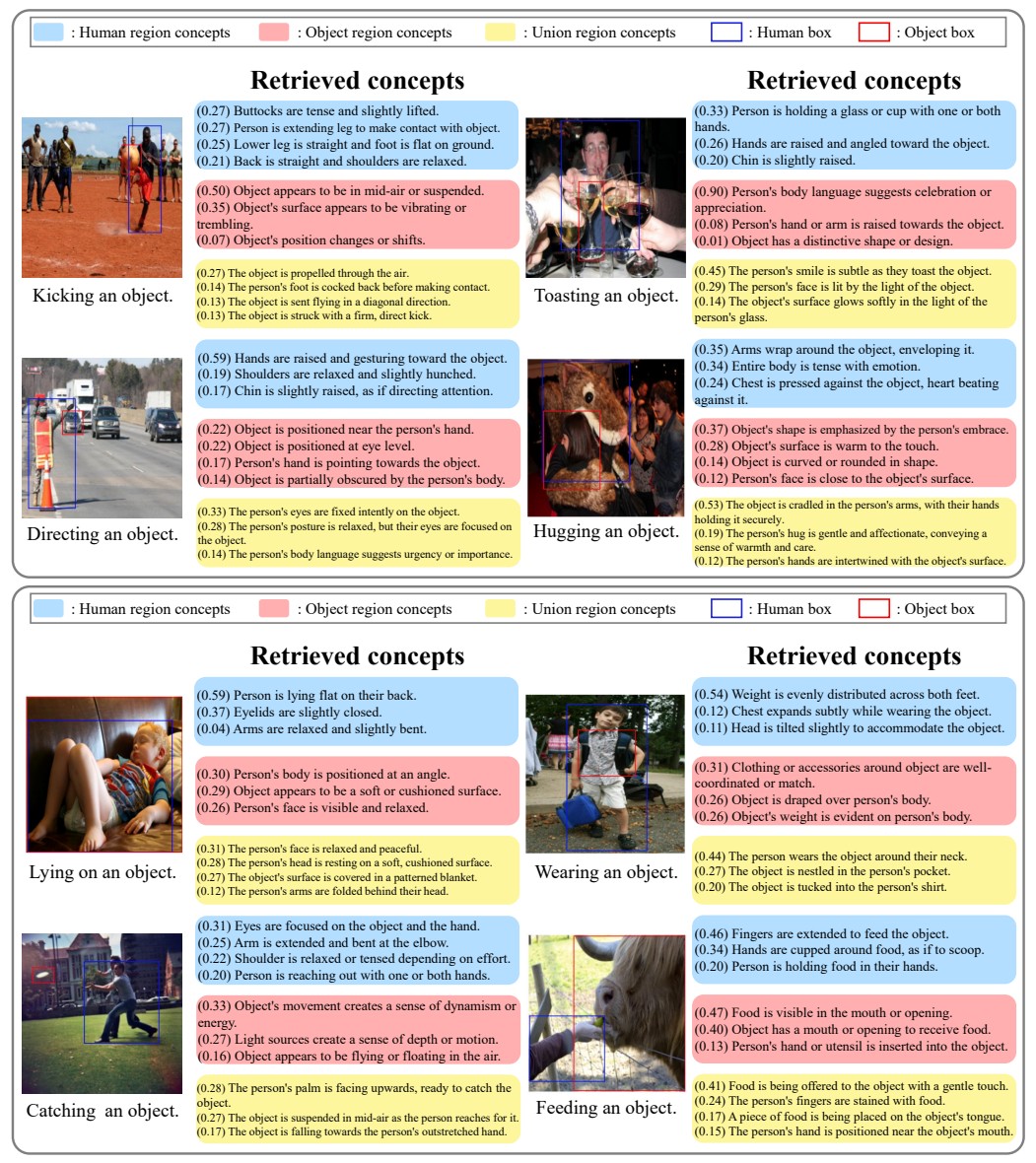

Figure 7: **Region-aware concept retrieval results (2/2).** Additional examples illustrating how different regions contribute to enhancing discriminability.

concept pool contains $K = 10$ entries per region type and verb class, and weights are displayed in parentheses. Due to sparsemax activation, concepts with low similarity scores are often assigned zero weight, resulting in more selective and interpretable retrieval. For instance, in *kicking an object*, the object region retrieves dynamic descriptions such as "(0.50) Object appears to be mid-air or suspended," while the human region emphasizes posture concepts like "(0.27) Buttocks are tense and slightly lifted." These complementary signals capture both motion and pose indicative of a kicking action. Similarly, in distinguishing *hugging* from *toasting*, human concepts like "(0.35) Arms wrap around the object" help identify close physical contact, whereas in toasting, object and union concepts highlight celebratory gestures such as "(0.90) Person's body language suggests celebration" and "(0.45) The person's smile is subtle as they toast the object." These examples show how our model selectively retrieves fine-grained region-specific concepts that support verb disambiguation in subtle interaction contexts. Overall, these results show that our region-aware prompt augmentation (RAP) selects meaningful concepts from distinct regions, supporting both interpretability and fine-grained HOI discrimination.

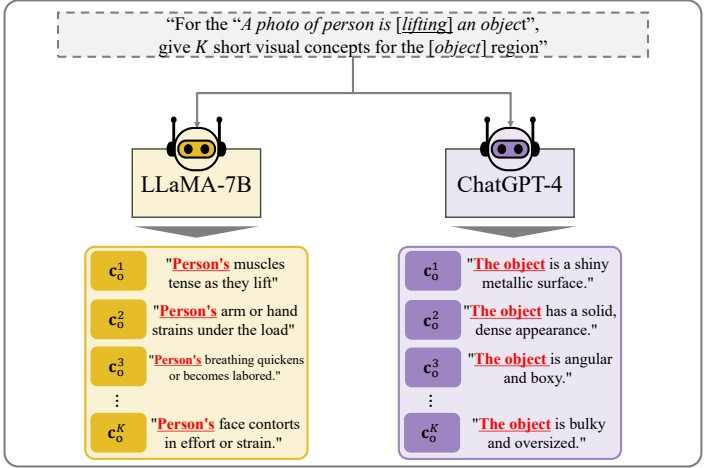

Figure 8: **Qualitative comparison of concept generation for the object region.** LLaMA-7B often yields human-centric concepts (left), while ChatGPT-4 provides more object-centered descriptions (right). This difference highlights the source of semantic noise in object-region concept retrieval.

Table 17: Comparison of performance when using object concepts from LLaMA-7B and ChatGPT-4 across four zero-shot settings. Best results are in **bold**.

| NF-UC | Full | Unseen | Seen | | RF-UC | Full | Unseen | Seen |
|---|---|---|---|---|---|---|---|---|
| LLaMA-7B | 32.18 | 36.16 | 31.18 | | LLaMA-7B | 33.55 | 31.24 | 34.13 |
| ChatGPT-4 | **32.57** | **36.45** | **31.60** | | ChatGPT-4 | **33.78** | **31.29** | **34.41** |

| UO | Full | Unseen | Seen | | UV | Full | Unseen | Seen |
|---|---|---|---|---|---|---|---|---|
| LLaMA-7B | 32.79 | 35.04 | 32.34 | | LLaMA-7B | **32.78** | **26.88** | **33.74** |
| ChatGPT-4 | **33.39** | **36.13** | **32.84** | | ChatGPT-4 | 32.73 | 26.69 | 33.72 |

## J   Analysis of object concept noise from LLaMA-7B

We further analyze the qualitative and quantitative effects of noisy object concepts generated by LLaMA-7B [48]. As discussed in the main paper, the prompt template *"a photo of a person [verb]-ing the object"* sometimes causes semantic leakage from the human region, producing human-centric descriptions for the object region. Fig. 8 visualizes this issue for the verb *lift*, where LLaMA-7B frequently associates object concepts with human body parts or actions. In contrast, ChatGPT-4 [49] produces more object-centered and visually grounded descriptions, effectively reducing such leakage. To quantitatively evaluate the impact of concept quality, we compare the model performance when object concepts are generated by LLaMA-7B versus ChatGPT-4. As shown in Table 17, GPT-4 generally achieves higher performance across all zero-shot settings, confirming that cleaner concept representations improve robustness without altering the model architecture.

