# OpenReview forum: "Visual Diversity and Region-aware Prompt Learning for Zero-shot HOI Detection"
_NeurIPS.cc/2025/Conference — NeurIPS 2025 poster_

### Official Review · Reviewer_7Lyn · 2025-06-29

**Clarity:** 3
**Significance:** 3
**Originality:** 2
**Rating:** 4
**Confidence:** 5

**Summary:**

This paper proposes a Visual Diversity and Region-aware Prompt Learning framework for zero-shot HOI detection, aiming to address the visual complexity inherent in human-object interactions. To this end, the authors introduce a visual diversity-aware prompt learning strategy that incorporates group-wise visual variance into the context embeddings. Experimental results on the HICO-DET benchmark validate the effectiveness of the proposed method.

**Questions:**

1. In Fig. 2, the paper adopts a fixed template — 'A photo of a person is verbing an object' — as the verb prompt. Did the authors explore alternative or automatically generated prompts? If not, it would be helpful to investigate whether different prompt formulations could impact performance.
2. The paper lacks an ablation study or empirical analysis to evaluate the contribution of Group-wise variance.
3. Although this paper is designed for zero-shot HOI detection, it can still be compared with other supervised methods. However, the authors did not conduct a comprehensive comparison.
4. The reported results for zero-shot HOI detection are not very competitive, and the paper lacks comparisons with recent state-of-the-art methods. For instance, SICHOI (CVPR 2024) reports a score of 36.27, and EZ-HOI (ECCV 2024) achieves 36.73 under the RF setting. Including such comparisons is essential to fairly evaluate the effectiveness of the proposed approach.
5. Since SICHOI also proposes the concept of multi-view image-text alignment, the novelty of the current work appears to be somewhat limited.

**Ethical Concerns:**

["NO or VERY MINOR ethics concerns only"]

**Final Justification:**

Thanks for the response from the authors. I increase my score from 3 to 4. However, I really think that the contribution of prompting design is not novel enough. The concept of region-aware prompting are not new in the community. Thus, I think the paper is marginally above the acceptance threshold.

**Quality:**

2

**Strengths And Weaknesses:**

Strengths: This paper proposes a Visual Diversity and Region-aware Prompt Learning framework for zero-shot HOI detection, aiming to address the visual complexity inherent in human-object interactions.. Moreover, it attempts to generate labels using a text-image model. The experimental results also confirm the effectiveness of the proposed algorithm.

Weaknesses: The paper aims to tackle the problem of zero-shot HOI detection. However, the proposed diversity-aware prompt learning strategy has already been explored in previous studies, limiting the novelty of the contribution. In addition, the experimental evaluation does not include a thorough comparison with current state-of-the-art methods, and the reported performance is not particularly competitive.

---

> ### Author Rebuttal · Authors · 2025-07-30
>
> We thank the reviewer for the thoughtful assessment and helpful suggestions. We address the reviewer’s concerns regarding novelty, ablation, and comparative evaluation below.
>
> ---
>
> ## 1. On the novelty of VDRP
>
> **As noted by reviewer FDPj, who highlighted “novel technical contributions” including our dual-module design and the variance injection and Sparsemax-based concept retrieval, our method introduces several key advances.**
>
> ### (1) Visual Diversity-aware Prompt (VDP)
>
> VDP provides a new perspective beyond prior distributional prompt methods such as ProDA and DAPT. While existing approaches operate in the text embedding space using non-parametric or regularization-based strategies, **VDP directly models prompt distributions from the visual modality by extracting union-region features per verb and injecting their group-wise variance into the prompt embedding.** This enables accurate modeling of intra-class diversity. Furthermore, unlike prior works that focus on seen class, VDP explicitly targets zero-shot HOI detection by transferring **visual variance from seen verbs to unseen ones via semantic grouping, improving generalization (Tab.5)**.
>
> ### (2) Region-Aware Prompt (RAP)
>
> Although both RAP and SICHOI explore multi-view alignment, their formulations differ in both scope and implementation:
> - **SICHOI** aligns **linguistic triplets (subject, verb, object)** with global image features using **dependency parsing**.
> - **RAP**, in contrast, aligns **visual regions (human, object, union)** with **region concepts**, refining prompts via **Sparsemax-based selection**.
>
> Our RAP introduces **region-specific, retrieval-based prompt augmentation**, offering a complementary and orthogonal design to SICHOI’s text-driven attention alignment.
>
> ---
>
> ## 2. On the ablation study and robustness of group-wise variance
>
> We respectfully disagree with the reviewer’s concern. **We have provided extensive analysis of our group-wise visual variance mechanism through ablations in both the main paper and supplementary.** Also, additional results are presented below for this rebuttal:
>
> ### 2.1 Analyses in the main paper and the supplement
>
> - **The main paper (Tab. 5)** compares **Global** vs. **Group_3** vs. **Group_5** grouping strategies.
> - **The supplement (Tab. 3)** evaluates the contribution of **mean + variance (μ, σ²)** vs. **variance-only (σ²)** to assess which statistics are most impactful.
> - These results confirm that **group-wise variance** improves robustness to visual diversity while being more stable than per-class estimation.
>
> ### 2.2 (New) Robustness to semantically isolated verbs
>
> To evaluate robustness under reduced semantic support, we **removed the top-3 semantic neighbors** for each unseen verb during variance computation:
>
> |  | **Full** | **Unseen** | **Seen** |
> | --- | --- | --- | --- |
> | W/O $\text{Top}-3$ | 32.64 | 26.21 | 33.69 |
> | W/ $\text{Top}-3$ | 32.78 | 26.88 | 33.74 |
>
> → Minimal performance drop (−0.67 mAP in unseen) confirms that our grouping retains generalization capacity even with limited support.
>
> ### 2.3 (New) Robustness to few-shot variance sampling
>
> We simulate low-data scenarios by limiting each verb class to **5 samples ($N_v$ = 5)** when estimating visual variance.
>
> | Setting | Sampling | Full | Unseen | Seen |
> | --- | --- | --- | --- | --- |
> | NF-UC | $N_v = 5$ | 32.23 | 35.90 | 31.31 |
> |  | All samples | 32.18 | 36.16 | 31.18 |
> | RF-UC | $N_v = 5$ | 33.40 | 30.59 | 34.10 |
> |  | All samples | 33.55 | 31.24 | 34.13 |
> | UO | $N_v = 5$ | 32.55 | 34.75 | 32.11 |
> |  | All samples | 32.79 | 35.04 | 32.34 |
> | UV | $N_v = 5$ | 32.89 | 25.88 | 34.04 |
> |  | All samples | 32.78 | 26.88 | 33.74 |
>
> → We observe only minor drops (≤ 0.2 full mAP), confirming that our grouping strategy produces stable variance estimates even with few samples.
>
> ---
>
> ## 4. On comparisons with SoTA (e.g., EZ-HOI, SICHOI)
>
> We thank the reviewer for pointing out the importance of comparison with recent SoTA methods such as EZ-HOI (NeurIPS 2024) and SICHOI (CVPR 2024). We provide a comprehensive and quantitative response below.
>
> ### 4.1 Comparison with EZ-HOI under all zero-shot settings
>
> **In the main paper and the supplement, we already compared** our method with **EZ-HOI (NeurIPS 2024)** across all four zero-shot HOI settings.
>
> **Our method consistently outperform EZ-HOI across all settings. Also,** note that the **reviewer’s reference to 36.73** corresponds to **Full mAP in the RF-UC**, which is already reported in our paper (Main: Table 1–3, Supplementary: Table 2).
> Under the same setting, our method achieves a higher **Full mAP of 38.13**, while also improving other key metrics such as **harmonic mean and unseen performance**.
>
> ### **(1) ResNet50+ViT-B**
>
> | Setting | Method | #TP | HM | Full | Unseen | Seen |
> | --- | --- | --- | --- | --- | --- | --- |
> | NF-UC | EZ-HOI | 6.85M | 31.76 | 31.17 | 33.66 | 30.55 |
> |  | **Ours** | **4.50M** | **33.49** | **32.18** | **36.16** | **31.18** |
> | RF-UC | EZ-HOI | 6.85M | 31.18 | 33.13 | 29.02 | 34.15 |
> |  | **Ours** | **4.50M** | **32.62** | **33.55** | **31.24** | **34.13** |
> | UO | EZ-HOI | 6.85M | 32.38 | 32.27 | 33.28 | 32.06 |
> |  | **Ours** | **4.50M** | **33.64** | **32.79** | **35.04** | **32.34** |
> | UV | EZ-HOI | 6.85M | 29.09 | 32.32 | 25.10 | 33.49 |
> |  | **Ours** | **4.50M** | **29.92** | **32.78** | **26.88** | **33.74** |
>
> ### **(2) ResNet50 + CLIP ViT-L**
>
> | **Setting** | **Method** | **#TP** | **HM** | **Full** | **Unseen** | **Seen** |
> | --- | --- | --- | --- | --- | --- | --- |
> | NF-UC | EZ-HOI | 14.07M | 35.38 | 34.84 | 36.33 | 34.47 |
> |  | **Ours** | **10.29M** | **36.83** | **36.46** | **37.48** | **36.21** |
> | RF-UC | EZ-HOI | 14.07M | 35.73 | 36.73 | 34.24 | 37.35 |
> |  | **Ours** | **10.29M** | **37.58** | **38.13** | **36.72** | **38.48** |
> | UO | EZ-HOI | 14.07M | 37.06 | 36.38 | 38.17 | 36.02 |
> |  | **Ours** | **10.29M** | **38.41** | **37.81** | **39.36** | **37.50** |
> | UV | EZ-HOI | 14.07M | 32.84 | 36.84 | 28.82 | 38.15 |
> |  | **Ours** | **10.29M** | **34.31** | **37.18** | **31.16** | **38.16** |
>
> These results demonstrate that our method achieves **state-of-the-art performance across all four zero-shot settings** and two backbones, consistently improving generalization capability to unseen interactions.
>
> > We also clarify that EZ-HOI is a NeurIPS 2024 paper (not ECCV). Our comparisons are consistent with the official ViT-B and ViT-L configurations reported in their NeurIPS version.
> >
>
> ---
>
> ### 4.2 Clarification on SICHOI and its limited comparability
>
> Thank you for sharing a recent and interesting work, **SICHOI (CVPR 2024).** We will **include this work in the final version.**  We acknowledge the recent advancement of SICHOI (CVPR 2024) performance. However, direct and fair comparison with our framework is problematic for several key reasons: **1)** **No code or model release, 2) No results on UO or UV settings, and more importantly 3) Unfairly high computational cost induced by a stronger backbone, additional foundation models (e.g., BLIP, SAM) and multiple decoders/branches.**
>
> | Model | Code? | UO/UV Reported? | External Modules |
> | --- | --- | --- | --- |
> | SICHOI | ✗ | ✗ | BLIP, SAM, CLIP |
> | Ours | ✓ | ✓ | CLIP |
>
> For reference, the **reviewer’s cited score of 36.27 corresponds to Unseen mAP under RF-UC**, while SICHOI also reports **36.44 under NF-UC**. We include both below for comparison.
>
> | Model | Backbone | RF-UC Unseen | NF-UC Unseen |
> | --- | --- | --- | --- |
> | SICHOI | **ResNet101** + ViT-L | 36.27 | 36.44 |
> | Ours | ResNet50 + ViT-L | **36.72** | **37.48** |
>
> **Also, supervised comparison is already provided in Supplementary Tab.1,** showing that our method performs competitively under the **supervised setting.**
>
> ---
>
> We hope these address the reviewer’s concern regarding comparison fairness and highlight the **superior generalization capability** of our approach across multiple zero-shot settings.
>
> ## 5. On the use of fixed prompt templates
>
> As reviewer requested, we experimented with three formulations:
>
> - **BASE template**: "A photo of a person is [verb]-ing an object."
> - **Simplified template**: "Person [verb] the object."
> - **LLM-style template**: "A example of the person [verb]-ing the object."
>
> Results show **only marginal differences**, indicating that our framework is **robust to the template design**:
>
> ### **NF-UC**
>
> | Template | **HM** | **Full** | **Unseen** | **Seen** |
> | --- | --- | --- | --- | --- |
> | a photo of a person [verb] + [ing] object (default) | 33.49 | 32.18 | 36.16 | 31.18 |
> | person [verb] the object | 33.65 | 32.40 | 36.17 | 31.46 |
> | a example of the person [verb] + [ing] object | 33.34 | 32.15 | 35.72 | 31.25 |
>
> ### **RF-UC**
>
> | Template | **HM** | **Full** | **Unseen** | **Seen** |
> | --- | --- | --- | --- | --- |
> | a photo of a person [verb] + [ing] object (default) | 32.62 | 33.55 | 31.24 | 34.13 |
> | person [verb] the object | 32.36 | 33.62 | 30.56 | 34.39 |
> | a example of the person [verb] + [ing] object | 31.59 | 33.08 | 29.52 | 33.97 |
>
> ### **UO**
>
> | Template | **HM** | **Full** | **Unseen** | **Seen** |
> | --- | --- | --- | --- | --- |
> | a photo of a person [verb] + [ing] object (default) | 33.64 | 32.79 | 35.04 | 32.34 |
> | person [verb] the object | 33.82 | 32.99 | 35.21 | 32.54 |
> | a example of the person [verb] + [ing] object | 33.57 | 32.83 | 34.77 | 32.45 |
>
> ### **UV**
>
> | Template | **HM** | **Full** | **Unseen** | **Seen** |
> | --- | --- | --- | --- | --- |
> | a photo of a person [verb] + [ing] object (default) | 29.92 | 32.78 | 26.88 | 33.74 |
> | person [verb] the object | 29.74 | 32.89 | 26.46 | 33.94 |
> | a example of the person [verb] + [ing] object | 29.95 | 33.05 | 26.70 | 34.08 |
>
> Template variation shows minimal performance impact, indicating that our prompt learning framework is robust to sentence structure.
>
> ---
>
> We truly appreciate the reviewer’s thoughtful comments and critical insights. We hope our responses and new analyses help resolve the concerns.

---

> > ### Comment · Reviewer_7Lyn · 2025-08-08
> > **Noverty of the VDRP**
> >
> > Thanks for the response from the authors. I am inclined to increase my score. However, I really think that the contribution of prompting design is not novel enough. The concept of region-aware prompting are not new in the community, though the results have proved its effectiveness. Thus, I think the paper is marginally above the acceptance threshold

---

> ### Author Response · Authors · 2025-08-08
>
> Dear Reviewer,
>
> We sincerely appreciate your careful review and the time you invested in evaluating our work.
> Your candid feedback has been valuable in helping us better understand how our contributions are perceived within the community.
> We are grateful that you considered our rebuttal and acknowledged the effectiveness of our approach, and we will take your suggestions into account to our paper.
>
> Best regards,
> Authors

---

### Official Review · Reviewer_2XwT · 2025-06-29

**Clarity:** 4
**Significance:** 3
**Originality:** 2
**Rating:** 5
**Confidence:** 4

**Summary:**

This paper presents VDRP (Visual Diversity and Region-aware Prompt learning), for zero-shot Human-Object Interaction detection. The work is motivated by two well-defined challenges inherent to the HOI task. The first is intra-class visual diversity, where a single verb class can be realized through a wide variety of visually distinct poses, scales, and contexts. The second challenge is inter-class visual entanglement, where semantically different verbs often produce visually similar human-object configurations, making them hard to disambiguate from global features alone.

To address these challenges, VDRP proposes two complementary modules that enhance the learnable textual prompts used for interaction classification within a frozen Vision-Language Model like CLIP.

1. Visual Diversity-aware Prompts (VDP): To tackle intra-class diversity, this module learns a distribution of prompts for each verb rather than a single static embedding. It first estimates the visual feature variance for each verb class from the training data. To create more robust estimates, especially for rare verbs, it groups semantically similar verbs and averages their variances. This "group-wise variance" is then used in two ways: it modulates the learnable context of the prompt via a small MLP, and it scales the magnitude of Gaussian noise that is added to the final prompt embedding. This process yields prompts that are designed to cover the diverse visual manifold of each verb class.

2. Region-aware Prompts (RAP): To resolve inter-class entanglement, this module augments the prompts with fine-grained, localized visual information. For each verb, it pre-generates a pool of descriptive textual "concepts" for the human, object, and union regions using an external Large Language Model (LLaMA-7B). During inference, the model dynamically retrieves the most relevant concepts from this pool by comparing them to the visual features extracted from the corresponding image regions (human, object, and union). These retrieved concepts are then added to the diversity-aware prompt, creating a final region-aware prompt that is sensitive to the subtle, localized cues necessary to distinguish between visually similar interactions.

The paper integrates these two modules into a unified, two-stage HOI detection pipeline. Extensive experiments on the HICO-DET benchmark demonstrate that VDRP achieves new state-of-the-art performance across four standard zero-shot settings, while being more parameter-efficient than previous methods.

**Questions:**

1. The VDP module is conceptually very similar to prior work in prompt distribution learning, such as ProDA[1] and DAPT[2], which also model distributions over prompts to capture visual variance. Could the authors please elaborate on the specific novel contributions of VDP beyond the application of these principles to the HOI detection task?

2. As the authors stated in the Supplementary material, the performance of the RAP module is implicitly dependent on the quality of the concepts generated by the external LLaMA-7B model. Has there been any analysis of the model's sensitivity to the quality of these concepts? For example, Table 6 shows that Softmax outperforms your chosen Sparsemax strategy in the UO setting, which seems to contradict the motivation of aggressively pruning irrelevant concepts. Could this difference suggest the concept pool is noisy, making Sparsemax's pruning harmful in some cases?

Overall, I believe this paper offers valuable insights into the longstanding issues of intra-class visual diversity and inter-class visual entanglement in the HOI detection task. My score currently falls between 4 and 5; if the authors can provide reasonable comments addressing my concerns, I am willing to raise my score.

References:

[1] Prompt Distribution Learning

[2] Distribution-Aware Prompt Tuning for Vision-Language Models

**Ethical Concerns:**

["NO or VERY MINOR ethics concerns only"]

**Final Justification:**

I appreciate the authors’ rebuttal.

I agree VDP is a well-designed module for the zero-shot HOI detection task. Although it builds upon existing methods such as ProDA and DAPT, the authors have demonstrated its effectiveness, particularly in zero-shot settings.

Regarding the robustness of the RAP module, I generally agree with the authors’ explanation that concept generation noise leads to performance variations, as evidenced by the τ-Sparsemax experiments. However, considering that HICO-DET contains only 600 HOI classes, maybe the authors should explore leveraging more advanced LLMs for visual concept generation to further enhance performance. In 2025, it is time to put LLaMA-7B out to pasture.

Based on the clarifications and the overall contribution, I will increase my score from 4 to 5.

**Limitations:**

yes

**Quality:**

3

**Strengths And Weaknesses:**

# Quality

## Strengths

The paper is well-motivated, grounding its contributions in a clear and compelling analysis of the core challenges in zero-shot HOI detection. The empirical evaluation is comprehensive and methodologically sound. The model presented in the paper achieved State-of-the-Art performance in all zero-shot settings on the HICO-DET dataset, with thorough ablation studies to validate the contributions of the proposed components.

## Weaknesses

As the author stated in the "Limitations" section in the Supplementary material, The RAP module's effectiveness is critically dependent on the quality and consistency of the concepts generated by the LLM. This introduces several potential points of failure and a lack of robustness that are not adequately addressed in the paper. LLMs are known to be susceptible to generating generic, biased, or factually incorrect information, and their outputs can vary based on the specific model version and prompting strategy. The paper provides no analysis of the quality of the generated concepts, nor does it describe any mechanism for filtering or verifying them. The performance of retrieval-augmented models is known to be sensitive to the quality of the retrieved information. Providing noisy or irrelevant context can actively harm model performance, a phenomenon well-documented in retrieval-augmented NLP. The RAP module is a form of retrieval-augmented generation, yet the paper does not evaluate its robustness to a noisy concept pool.

# Clarity

The paper is well-written, clearly structured, and easy to follow. The authors have done an excellent job of motivating the problem and presenting their solution.

# Significance

## Strengths

The work makes a significant and timely contribution to the field. Zero-shot HOI detection is a critical and challenging task for advancing scene understanding, and this paper pushes the state-of-the-art forward meaningfully.

## Weaknesses

However, the problem of intra-class visual diversity and inter-class visual entanglement has been identified by some previous works[1][2], although this paper proposed a meaningful method to improve the performance on this task, it is not the first paper to contribute on these issues.

# Originality

The originality of the paper is good, but it is the area with the most notable weakness. The novelty lies more in the successful synthesis and specific adaptation of existing ideas to the HOI domain, rather than the invention of fundamentally new techniques.

The core idea of the VDP module is learning a distribution over prompts to capture the visual variance of a class, which is a well-established method. Paper "Prompt Distribution Learning" from CVPR 2022, explicitly proposes learning a Gaussian distribution over prompt embeddings to handle intra-class visual diversity in few-shot classification, also, I suggest the authors to cite this paper. The VDP module's use of variance to modulate a context vector and to scale Gaussian noise applied to the final prompt embedding is a direct application of this established principle. The main novel element here appears to be the "group-wise variance estimation", which is a clever engineering solution to handle data sparsity for rare verbs, but the conceptual mechanism of learning a prompt distribution is inherited. However, I must emphasize that the VDP module is well tailored for the HOI detection task.

References:

[1] Bongard-HOI: Benchmarking Few-Shot Visual Reasoning for Human-Object Interactions

[2] Open-World Human-Object Interaction Detection via Multi-modal Prompts

---

> ### Author Rebuttal · Authors · 2025-07-30
>
> We thank the reviewer for the constructive feedback. We appreciate the recognition of our motivation and its effectiveness in addressing intra-class diversity and inter-class entanglement in HOI detection. Below, we clarify the novelty of VDP and the robustness of RAP with supporting analyses.
>
> ---
>
> ### 1. On the novelty and contributions of VDP
>
> We thank the reviewer for pointing out connections to prior distribution-based prompt learning methods such as ProDA and DAPT.
> While all three methods adopt a distributional approach, VDP introduces several key innovations that make it uniquely suited for zero-shot HOI detection.
>
> **Importantly, as noted by Reviewer FDPj, our design offers “novel technical contributions”, and both variance injection and Sparsemax-based concept retrieval are considered innovative**.
>
> ### (1) Difference in distribution modeling strategy
>
> Unlike ProDA and DAPT, which model prompt distributions through non-parametric or regularization-based methods on text embeddings, **VDP directly leverages visual modality** to guide the prompt distribution.
> Specifically, we extract **union-region visual features per verb** and **inject their group-wise variance** into the context vectors of the prompt learner. This method enables more faithful modeling of intra-class diversity in HOI representations.
>
> ### (2) Zero-shot verb generalization
>
> Both ProDA and DAPT focus on seen-class classification tasks.
> In contrast, **VDP is explicitly designed for zero-shot HOI detection**, with variance modeling applied to unseen verb classes via **semantically grouped visual variance transfer**, as shown in **Table 5 (Group-5 > Global)**.
> This structure-aware grouping stabilizes learning and improves generalization to rare or novel classes.
>
> ### (3) Region-aware extension
>
> VDP is also integrated into a dual-module architecture (VDP + RAP), where **RAP introduces region-specific augmentation via Sparsemax-based concept selection** from human, object, and union regions.
> This region-aware design is absent in prior works and further improves verb disambiguation by capturing localized concepts.
>
> ### (4) Structured prompt distribution without explicit regularization
>
> Unlike methods such as CMMP that rely on explicit loss functions to shape the prompt space, **VDP naturally induces a structured and balanced distribution** through variance-guided training.
> Based on the analysis we conducted in **Supplementary Section 6 and Fig. 1**, prompt embeddings evolve from a collapsed space (**0.105 ± 0.031**) to a more expressive and semantically structured representation (**0.550 ± 0.103**), **closely reflecting the diversity of visual features (0.518 ± 0.105)**.
> In contrast, CMMP over-expands the prompt space (0.937 ± 0.133), resulting in loss of semantic structure.
> These results highlight that VDP produces **well-regularized, diversity-aware prompts** without requiring additional regularization losses.
>
> ---
>
> ### 2. On the robustness and sensitivity of RAP to concept quality
>
> While we acknowledge that RAP’s performance can be affected by the quality of LLM-generated concepts, **our analysis shows that the Sparsemax strategy is generally robust.** The slight drop in UO performance is not due to Sparsemax pruning itself, but rather to semantic noise arising from imperfect concept generation and region-feature alignment.
>
> ### 2.1 Qualitative diagnosis of noisy object concepts
>
> The real source of noise lies in **semantic leakage from the human region** during concept generation. Since our prompts follow the template “a photo of a person [verb]-ing **the object**”, LLMs often generate **object concepts with unintended human descriptions.**
>
> For example, object concept pool for “lift” includes:
>
> > "Person's legs are crossed or positioned comfortably.", "Person's feet are on the ground or resting on the object.",  "Person's arms are folded or resting on their lap.", etc.
> >
>
> These human-centric concepts are particularly problematic in **object concepts retrieval**, because **object region crops frequently include parts of the human (e.g., hands, arms)** due to physical proximity. This causes the region feature to match **human-like concepts**, leading Sparsemax to assign them higher weights—**pruning away actual object-relevant concepts**.
>
> ### 2.3 Retrieval strategy sensitivity: τ-Sparsemax
>
> We further test the effect of adaptive sparsity by applying τ-Sparsemax (τ ∈ {0.0, 0.05, 0.1}), where weights below τ are zeroed. As shown below, **our model remains robust across all τ values**, with only minor performance fluctuations.
>
> | τ value | **HM** | **Full** | **Unseen** | **Seen** |
> | --- | --- | --- | --- | --- |
> | 0.00 (default) | 33.49 | 32.18 | 36.16 | 31.18 |
> | 0.05 | 33.51 | 32.38 | 35.76 | 31.53 |
> | 0.10 | 33.44 | 32.23 | 35.85 | 31.33 |
>
> ### **RF-UC**
>
> | τ value | **HM** | **Full** | **Unseen** | **Seen** |
> | --- | --- | --- | --- | --- |
> | 0.00 (default) | 32.62 | 33.55 | 31.24 | 34.13 |
> | 0.05 | 31.62 | 33.10 | 29.56 | 33.99 |
> | 0.10 | 32.50 | 33.59 | 30.92 | 34.26 |
>
> ### **UO**
>
> | τ value | **HM** | **Full** | **Unseen** | **Seen** |
> | --- | --- | --- | --- | --- |
> | 0.00 (default) | 33.64 | 32.79 | 35.04 | 32.34 |
> | 0.05 | 33.59 | 32.95 | 34.63 | 32.62 |
> | 0.10 | 33.98 | 32.90 | 35.84 | 32.31 |
>
> ### **UV**
>
> | τ value | **HM** | **Full** | **Unseen** | **Seen** |
> | --- | --- | --- | --- | --- |
> | 0.00 (default) | 29.92 | 32.78 | 26.88 | 33.74 |
> | 0.05 | 29.35 | 32.73 | 25.91 | 33.84 |
> | 0.10 | **29.99** | **32.85** | **26.95** | **33.81** |
>
> These results demonstrate that **RAP is robust to sparsity thresholds**, and that performance variations are more likely caused by **concept generation noise**, not the retrieval strategy itself.
>
> ---
>
> ### 3. Clarification on the positioning relative to prior work
>
> We acknowledge that the **challenges of intra-class diversity and inter-class entanglement** have been previously recognized (e.g., Bongard-HOI, Open-World HOI). However, our contribution lies in:
>
> - Structuring these challenges into a **unified dual-module framework** (VDP + RAP),
> - Providing a **visually grounded solution** that addresses both variation and confusion explicitly through prompt learning,
> - And validating it under four **zero-shot settings** on HICO-DET with **consistent SOTA performance and low parameter count (4.5M)**.
>
> Thus, while the problem is not entirely new, **our solution is novel in formulation, integration, and empirical strength**.
>
> ---
>
> ### 4. Citations and acknowledgements
> We thank the reviewer for suggesting citations to ProDA and DAPT. **We will incorporate them in the revised version and clarify how our formulation builds upon and differs from these prior works.**
>
> ---
>
> We sincerely thank the reviewer for pointing out the sensitivity of RAP to concept quality and the connection between VDP and prior prompt distribution methods. We hope our clarifications and analyses help address these concerns.

---

> ### Author Response · Authors · 2025-08-06
> **Follow-up on rebuttal response**
>
> Dear Reviewer,
>
> Thank you for acknowledging our rebuttal. We hope it addressed your main concerns—particularly regarding contributions of VDP, sensitivity of RAP.
>
> If there are any remaining questions or suggestions, we would be more than happy to clarify them during the discussion phase.
>
> We sincerely appreciate your time and thoughtful review.
>
> Best regards,
> Authors

---

### Official Review · Reviewer_cCr1 · 2025-06-30

**Clarity:** 3
**Significance:** 3
**Originality:** 3
**Rating:** 4
**Confidence:** 5

**Summary:**

This paper proposes VDRP, a framework specifically designed to enhance Zero-shot Human-Object Interaction (HOI) detection by addressing two primary challenges: intra-class visual diversity and inter-class visual entanglement. VDRP employs a visual diversity-aware prompt learning strategy by injecting group-wise visual variations into prompts and applying Gaussian perturbation to capture diverse verb visualizations. Additionally, it incorporates region-specific embeddings from human, object, and union regions to generate region-aware prompts with LLM, thus improving verb-level discrimination. Experimental evaluation on the HICO-DET benchmark shows that VDRP achieves SOTA results on the zero-shot settings of HICO-DET.

**Questions:**

In Line 150, the mean $\mu_v$ and variance $\sigma_v^2$ are computed over all $N_v$ samples, what if the $N_v$ is fixed to a small number, e.g., 5? Will the performance be affected significantly?

After training, what is the distribution of the diversity-aware prompt embeddings $\tilde{t}^v$? What is the different between the original token embeddings $\bar{P}_v$?

**Ethical Concerns:**

["NO or VERY MINOR ethics concerns only"]

**Final Justification:**

Thank you for the authors' efforts and for providing detailed supplementary results. Based on the additional results, I observe that the choice of the variance injection parameter $\alpha$ only marginally affects the outcomes. The selection of the concept scaling factor $\beta$ does not have a significant impact on the results, except in the NF-UC setting. Additionally, the improvement in accuracy on unseen samples achieved by Gaussian perturbation aligns well with expectations.

Regarding the zero-shot results with $N_v=5$, increasing the number of samples appears to have limited effect. Overall, these additional results have addressed my previous concerns. However, I would like to note that most of the observed improvements are not statistically significant. Therefore, I will maintain my original score.

**Limitations:**

Yes.

**Paper Formatting Concerns:**

None.

**Quality:**

3

**Strengths And Weaknesses:**

Strengths:

This paper explores the intra-class visual information for zero-shot HOI detection. The proposed visual diversity-aware prompt learning strategy facilitates the verb co-occurrence representation of the verb embeddings. The retrieval-based region prompt augmentation, an advanced LLM guided prompt refinement method, fuses the region-level embeddings while filtering out the irrelevant concepts in the verb embeddings. The proposed method achieves SOTA results with comparable efficiency to the previous methods on the zero-shot settings of HICO-DET.

Weaknesses:

Both the VDP and RAP is carefully designed for specific purposes, however, the effectiveness of some important part of the introduced components are not sufficient. In Eq. (8), adjusting the hyper-parameter $\alpha$ directly affects the contribution of the proposed group-wise variance, the impact of the variance is not clear. Similarly, the hyper-parameter $\beta$ in Eq. (13). In addition, the contribution of the Gaussian perturbation (with or without) according to recent findings is not verified in this paper.

---

> ### Author Rebuttal · Authors · 2025-07-30
>
> We thank the reviewer for the thoughtful feedback and the recognition of our contributions to modeling intra-class visual diversity and inter-class entanglement in zero-shot HOI detection. Below, we address all the questions/concerns raised by the reviewer.
>
> ---
>
> ### 1. Analysis of hyperparameters (α, β, Gaussian perturbation)
>
> As requested, we provide additional analyses of hyperparameters across four zero-shot settings:
>
> - **α** (Eq. 8): scaling factor for the delta vector derived from group-wise variance.
> - **β** (Eq. 13): scaling factor for region-aware concepts augmentation.
> - **Contribution for Gaussian perturbation**
>
> **Our analyses show that the proposed method is robust to the choice of hyperparameters in all settings and Gaussian perturbation added prompt embedding improves generalization capability in “unseen” classes.**
>
> ### 1.1 Impact of variance injection scale (α)
>
> We conduct ablations on α ∈ {0.02, 0.01, 0.1} to assess the effect of delta scaling in Eq. (8). The results are summarized below:
>
> ### **NF-UC**
>
> | α | **HM** | **Full** | **Unseen** | **Seen** |
> | --- | --- | --- | --- | --- |
> | 0.02 (default) | 33.49 | 32.18 | 36.16 | 31.18 |
> | 0.01 | 33.13 | 32.03 | 35.29 | 31.22 |
> | 0.1 | 33.27 | 32.02 | 35.80 | 31.07 |
>
> ### **RF-UC**
>
> | α | **HM** | **Full** | **Unseen** | **Seen** |
> | --- | --- | --- | --- | --- |
> | 0.02 (default) | 32.62 | 33.55 | 31.24 | 34.13 |
> | 0.01 | 32.31 | 33.51 | 30.58 | 34.24 |
> | 0.1 | 31.79 | 33.16 | 29.86 | 33.98 |
>
> ### **UO**
>
> | α | **HM** | **Full** | **Unseen** | **Seen** |
> | --- | --- | --- | --- | --- |
> | 0.02 (default) | 33.64 | 32.79 | 35.04 | 32.34 |
> | 0.01 | 33.82 | 32.81 | 35.52 | 32.27 |
> | 0.1 | 33.67 | 32.81 | 35.11 | 32.35 |
>
> ### **UV**
>
> | α | **HM** | **Full** | **Unseen** | **Seen** |
> | --- | --- | --- | --- | --- |
> | 0.02 (default) | 29.92 | 32.78 | 26.88 | 33.74 |
> | 0.01 | 29.27 | 32.48 | 25.97 | 33.54 |
> | 0.1 | 28.57 | 32.54 | 24.73 | 33.81 |
>
> We find that α = 0.02 offers the best trade-off across most settings. This value (0.02) was chosen to match the initialization scale of CLIP’s context embeddings, which ensures stability as large delta scales can lead to over-perturbed context tokens and unstable training. Since context tokens are known to be sensitive to initialization, keeping the scale aligned enhances stability while capturing diversity.
>
> ### 1.2 Concepts scaling factor (β)
>
> We evaluate β ∈ {0.2, 0.5, 1.0} in Eq. (13) to study how strongly region-derived concepts should contribute to the prompt augmentation.
>
> The results are summarized below:
>
> ### **NF-UC**
>
> | β | **HM** | **Full** | **Unseen** | **Seen** |
> | --- | --- | --- | --- | --- |
> | 0.2 (default) | 33.49 | 32.18 | 36.16 | 31.18 |
> | 0.5 | 33.39 | 32.14 | 35.92 | 31.19 |
> | 1.0 | 32.96 | 31.85 | 35.15 | 31.02 |
>
> ### **RF-UC**
>
> | β | **HM** | **Full** | **Unseen** | **Seen** |
> | --- | --- | --- | --- | --- |
> | 0.2 (default) | 32.62 | 33.55 | 31.24 | 34.13 |
> | 0.5 | 32.38 | 33.52 | 30.73 | 34.21 |
> | 1.0 | 32.33 | 33.48 | 30.67 | 34.18 |
>
> ### **UO**
>
> | β | **HM** | **Full** | **Unseen** | **Seen** |
> | --- | --- | --- | --- | --- |
> | 0.2 (default) | 33.64 | 32.79 | 35.04 | 32.34 |
> | 0.5 | 33.87 | 33.08 | 35.17 | 32.66 |
> | 1.0 | 33.41 | 32.60 | 34.74 | 32.17 |
> |  |  |  |  |  |
>
> ### **UV**
>
> | β | **HM** | **Full** | **Unseen** | **Seen** |
> | --- | --- | --- | --- | --- |
> | 0.2 (default) | 29.92 | 32.78 | 26.88 | 33.74 |
> | 0.5 | 29.99 | 32.72 | 27.06 | 33.64 |
> | 1.0 | 29.27 | 32.54 | 25.92 | 33.62 |
>
> Moderate augmentation scaling (β = 0.2) achieves the best balance overall. Excessive weighting (β = 1.0) slightly degrades performance across multiple settings, due to overconfident region concepts.
>
> ### 1.3 Effect of Gaussian perturbation (ε)
>
> We evaluate the effect of adding stochasticity to prompt embeddings via ε ∈ {0, 0.1}. The results below demonstrate that the Gaussian perturbation consistently improves generalization capability in **“unseen” classes.**
>
> ### **NF-UC**
>
> | Gaussian perturbation | **HM** | **Full** | **Unseen** | **Seen** |
> | --- | --- | --- | --- | --- |
> | W/O perturbation | 33.57 | 32.49 | 35.68 | 31.69 |
> | W/ perturbation (default) | 33.49 | 32.18 | **36.16** | 31.18 |
>
> ### **RF-UC**
>
> | Gaussian perturbation | **HM** | **Full** | **Unseen** | **Seen** |
> | --- | --- | --- | --- | --- |
> | W/O perturbation | 31.76 | 33.12 | 29.83 | 33.95 |
> | W/ perturbation (default) | 32.62 | 33.55 | **31.24** | 34.13 |
>
> ### **UO**
>
> | Gaussian perturbation | **HM** | **Full** | **Unseen** | **Seen** |
> | --- | --- | --- | --- | --- |
> | W/O perturbation | 33.67 | 33.09 | 34.59 | 32.79 |
> | W/ perturbation (default) | 33.64 | 32.79 | **35.04** | 32.34 |
>
> ### **UV**
>
> | Gaussian perturbation | **HM** | **Full** | **Unseen** | **Seen** |
> | --- | --- | --- | --- | --- |
> | W/O perturbation | 29.49 | 32.72 | 26.16 | 33.79 |
> | W/ perturbation (default) | 29.92 | 32.78 | **26.88** | 33.74 |
>
> Perturbation improves generalization in most unseen settings, confirming that stochastic sampling helps model diverse visual realizations for each verb.
>
> ---
>
> ### 2. Robustness of visual variance under few-shot sampling
>
> As requested, we provide additional analysis on the robustness of our model across four zero-shot settings. We estimate group-wise variance **using only 5 visual features per verb ($N_v = 5$)** and compare it to the default setting. Performance remains stable, indicating that our method generalizes well even under few-sample conditions.
>
> ### **NF-UC**
>
> | Sampling | **HM** | **Full** | **Unseen** | **Seen** |
> | --- | --- | --- | --- | --- |
> | Few-shot ($N_v = 5$) | 33.45 | 32.23 | 35.90 | 31.31 |
> | All samples (default) | 33.49 | 32.18 | 36.16 | 31.18 |
>
> ### **RF-UC**
>
> | Sampling | **HM** | **Full** | **Unseen** | **Seen** |
> | --- | --- | --- | --- | --- |
> | Few-shot ($N_v = 5$) | 32.25 | 33.40 | 30.59 | 34.10 |
> | All samples (default) | 32.62 | 33.55 | 31.24 | 34.13 |
>
> ### **UO**
>
> | Sampling | **HM** | **Full** | **Unseen** | **Seen** |
> | --- | --- | --- | --- | --- |
> | Few-shot ($N_v = 5$) | 33.38 | 32.55 | 34.75 | 32.11 |
> | All samples (default) | 33.64 | 32.79 | 35.04 | 32.34 |
>
> ### **UV**
>
> | Sampling | **HM** | **Full** | **Unseen** | **Seen** |
> | --- | --- | --- | --- | --- |
> | Few-shot ($N_v = 5$) | 29.40 | 32.89 | 25.88 | 34.04 |
> | All samples (default) | 29.92 | 32.78 | 26.88 | 33.74 |
>
> We observe only minor drops (≤ 0.2 full mAP), confirming that our grouping strategy produces stable variance estimates even with few samples.
>
> ---
>
> ### 3. Distribution of diversity-aware prompt
>
> **As requested, we analyze the distribution of prompt embeddings after training using the inter-class cosine distance metric.**
> This analysis was **conducted and reported in Section 6 of the Supplementary**, along with visualizations in Supplementary Fig.1.
> Prompt embeddings begin in a collapsed state (**0.105 ± 0.031**) and become significantly more diverse after training (**0.550 ± 0.103**), closely matching the visual space (**0.518 ± 0.105**).
> Unlike CMMP (**0.937 ± 0.133**), which over-separates classes, our method yields **structured and aligned distributions** that reflect visual diversity.
> These results confirm that our prompts evolve into meaningful representations without needing additional alignment losses.
>
> ---
>
> We sincerely thank the reviewer for raising insightful questions regarding the effect of group-wise variance (Eq. 8, 13) and the role of Gaussian perturbation in our framework. We hope our additional analyses and clarifications help address these concerns.

---

> ### Author Response · Authors · 2025-08-06
> **Follow-up on rebuttal response**
>
> Dear Reviewer,
>
> Thank you for your detailed review. We wanted to kindly follow up and check whether our rebuttal addressed your concerns.
>
> If any points remain unclear or if you have further suggestions, we would be happy to provide additional clarification during the discussion period.
>
> We sincerely appreciate your time and consideration.
>
> Best regards,
> Authors

---

### Official Review · Reviewer_FDPj · 2025-07-02

**Clarity:** 4
**Significance:** 3
**Originality:** 3
**Rating:** 4
**Confidence:** 4

**Summary:**

This paper proposes VDRP, a novel framework for zero-shot Human-Object Interaction (HOI) detection. The method addresses two key challenges: (1) intra-class visual diversity via visual diversity-aware prompts (VDP) that inject group-wise variance and Gaussian noise into CLIP prompts, and (2) inter-class visual entanglement (distinct verbs appear similar) via region-aware prompts (RAP) that augment prompts with LLM-generated concepts retrieved from human/object/union regions. VDRP achieves SOTA results with only 4.5M trainable parameters.

**Questions:**

1) Zero-Shot Verb Variance Robustness
Suggestion: Add experiments with intentionally isolated verbs (e.g., remove top-3 semantic neighbors from training). Report mAP on these verbs under UV setting.

2) Sparsemax Failure Diagnosis
Suggestion:
(a) Quantify concept noise: Report % of shared concepts across verbs (e.g., "table" in verb pairs) for UO vs. UV.
(b) Test adaptive thresholds: Replace Sparsemax with τ-Sparsemax (adjustable sparsity).

3) Sensitivity Issues of Sparsemax:
Table 6 shows that Softmax outperforms Sparsemax under the UO setting. The authors attribute this to “unreliable LLM concepts,” but no further analysis is provided:
(1) Is this due to higher concept noise in object regions (e.g., “table” appears in both “sit at” and “work on”)?
(2) Is the threshold selection sensitive? The authors did not attempt to adjust the sparsity level of Sparsemax.

**Ethical Concerns:**

["NO or VERY MINOR ethics concerns only"]

**Final Justification:**

The rebuttal partially clarifies certain issues, but my original evaluation persists after full consideration of all feedback.

**Limitations:**

1. **Computational costs**: No analysis of extra overhead from LLaMA queries/variance computation.
2. **Bias propagation**: LLM-generated concepts may encode societal biases (e.g., "woman cooking" for "use stove").
3. **Failure cases**: No discussion of scenarios where VDRP fails (e.g., heavy occlusion or rare verbs).

**Paper Formatting Concerns:**

The paper's formatting basically meets the requirements.

**Quality:**

4

**Strengths And Weaknesses:**

**Strengths**
1) Novel technical contributions: The dual-module design (VDP+RAP) elegantly solves core HOI challenges. Variance injection and Sparsemax-based concept retrieval are innovative.
2) Rigorous evaluation: Extensive experiments across all four standard zero-shot HOI settings with SOTA results.
3) High efficiency: Achieve SOTA with lightweight model.
4) Well-structured analysis: Ablations (Tables 4-6) and visualizations (Figures 1-4) effectively validate design choices.


**Weaknesses**
1) Fragile Zero-Shot Verb Generalization: Group-wise variance estimation (Eq 6) assumes training-set verbs adequately cover all semantic neighbors. For truly novel verbs with no similar training-set counterparts (e.g., "teeter"), variance degenerates to a global mean, failing to model diversity.
2) Sparsemax Sensitivity Unexplored: While Sparsemax aids concept filtering, its threshold sensitivity harms robustness in object-centric settings (UO in Table 6). No analysis confirms if failure stems from noisy object concepts (e.g., "table" for both "sit" and "work") or improper sparsity calibration.
3) CLIP Phrase Encoding Bias: The text encoder handles compound phrases ("hold+baseball+glove") less effectively than atomic concepts, introducing prompt bias. The fixed template design lacks ablation against alternative formulations (e.g., "person hold glove").

---

> ### Author Rebuttal · Authors · 2025-07-30
>
> We thank the reviewer for the constructive feedback. We are encouraged by the positive assessment of our technical contributions, evaluation rigor, and clarity. Below, we address the main concerns regarding zero-shot verb generalization, Sparsemax sensitivity, and phrase encoding bias.
>
> ---
>
> ### **1. Robustness to semantically isolated verbs**
>
> We thank the reviewer for highlighting a potential limitation of our group-wise variance estimation when applied to **semantically isolated unseen verbs**.
> To simulate this extreme case, we conduct an additional experiment under the UV setting, where we remove the top-3 semantic neighbors (based on CLIP text embedding similarity) for each unseen verb prior to computing group-wise visual variance. This setting emulates scenarios where unseen verbs lack nearby counterparts in the training set.
> As shown below, the unseen performance drops only slightly (−0.67 Unseen mAP), indicating that **our method maintains generalization ability even under reduced semantic support**:
>
> |  | **HM** | **Full** | **Unseen** | **Seen** |
> | --- | --- | --- | --- | --- |
> | W/O $\text{Top}-3$ | 29.48 | 32.64 | 26.21 | 33.69 |
> | W/ $\text{Top}-3$ | 29.92 | 32.78 | 26.88 | 33.74 |
>
> Also, our main paper also includes a related analysis using global variance estimation (Main paper: Tab.4), further supporting the robustness of our approach when class-specific support is limited.
>
> ---
>
> ### 2. Sparsemax failure diagnosis
>
> We confirm that our method is **robust to concept overlap and sparsity thresholding**, as validated through additional quantitative and ablation analyses.
>
> ### 2.1 Object concept overlap analysis
>
> To evaluate the reviewer’s hypothesis on concept noise due to shared object semantics (e.g., “table” in both “sit” and “work”), we compute the percentage of exact string overlaps in object concepts across verb pairs. Using cosine similarity ≥ 0.90, we find:
>
> - **UO setting**: 2.98% shared concepts
> - **UV setting**: 2.94% shared concepts
>
> These results suggest that **both settings exhibit similarly low overlap**, and the slightly lower UO performance appears to stem from stochastic variation rather than concept redundancy.
>
> ### **2.3 Qualitative diagnosis of noisy object concepts**
>
> While concept overlap remains low, we observe that some object concepts contain **human-centric descriptions** due to the prompt template (“a photo of a person [verb]-ing **the object**”) and lack of explicit object grounding. For example, object concepts for "lift" include:
>
> > "Person's legs are crossed or positioned comfortably.", "Person's feet are on the ground or resting on the object.", "Person's arms are folded or resting on their lap.", etc.
> >
>
> This suggests that **semantic leakage from the human region may introduce localized noise**, particularly in the UO setting. However, this appears to be a **minor stochastic effect** rather than a structural flaw, as overall performance remains stable.
>
> ### 2.3 τ-Sparsemax analysis
>
> We further test the effect of adaptive sparsity by applying τ-Sparsemax (τ ∈ {0.0, 0.05, 0.1}), where weights below τ are zeroed. As shown below, **our model remains robust across all τ values**, with only minor performance fluctuations.
>
> ### **NF-UC**
>
> | τ value | **HM** | **Full** | **Unseen** | **Seen** |
> | --- | --- | --- | --- | --- |
> | 0.00 (default) | 33.49 | 32.18 | 36.16 | 31.18 |
> | 0.05 | 33.51 | 32.38 | 35.76 | 31.53 |
> | 0.10 | 33.44 | 32.23 | 35.85 | 31.33 |
>
> ### **RF-UC**
>
> | τ value | **HM** | **Full** | **Unseen** | **Seen** |
> | --- | --- | --- | --- | --- |
> | 0.00 (default) | 32.62 | 33.55 | 31.24 | 34.13 |
> | 0.05 | 31.62 | 33.10 | 29.56 | 33.99 |
> | 0.10 | 32.50 | 33.59 | 30.92 | 34.26 |
>
> ### **UO**
>
> | τ value | **HM** | **Full** | **Unseen** | **Seen** |
> | --- | --- | --- | --- | --- |
> | 0.00 (default) | 33.64 | 32.79 | 35.04 | 32.34 |
> | 0.05 | 33.59 | 32.95 | 34.63 | 32.62 |
> | 0.10 | 33.98 | 32.90 | 35.84 | 32.31 |
>
> ### **UV**
>
> | τ value | **HM** | **Full** | **Unseen** | **Seen** |
> | --- | --- | --- | --- | --- |
> | 0.00 (default) | 29.92 | 32.78 | 26.88 | 33.74 |
> | 0.05 | 29.35 | 32.73 | 25.91 | 33.84 |
> | 0.10 | 29.99** | 32.85 | 26.95 | 33.81 |
>
> These results demonstrate that **RAP is robust to sparsity thresholds**, and that performance variations are more likely caused by **concept generation noise**, not the retrieval strategy itself.
>
> ---
>
> ### 3. CLIP phrase encoding bias
>
> We thank the reviewer for pointing out the potential bias in CLIP’s phrase encoding, especially for compound expressions like “holding+baseball+glove.”
> **While this is a known limitation of CLIP, our prompt template—“a photo of a person [verb]-ing the object”—is a standard formulation widely used in HOI literature.**
> To assess its effect, we performed an ablation study using alternative templates (e.g., “person [verb] the object”).
>
> Results show minimal performance difference, indicating that **our method is robust to such phrasing variations.**
> Specifically, we compare:
>
> - **BASE template**: "A photo of a person is [verb]-ing the object."
> - **Simplified template**: "Person [verb] the object."
> - **LLM-style template**: "A example of the person [verb]-ing the object."
>
> Results are summarized below across all four zero-shot settings:
>
> ### **NF-UC**
>
> | Template | **HM** | **Full** | **Unseen** | **Seen** |
> | --- | --- | --- | --- | --- |
> | a photo of a person [verb] + [ing] the object (default) | 33.49 | 32.18 | 36.16 | 31.18 |
> | person [verb] the object | 33.65 | 32.40 | 36.17 | 31.46 |
> | an example of the person [verb] + [ing] the object | 33.34 | 32.15 | 35.72 | 31.25 |
>
> ### **RF-UC**
>
> | Template | **HM** | **Full** | **Unseen** | **Seen** |
> | --- | --- | --- | --- | --- |
> | a photo of a person [verb] + [ing] the object (default) | 32.62 | 33.55 | 31.24 | 34.13 |
> | person [verb] the object | 32.36 | 33.62 | 30.56 | 34.39 |
> | an example of the person [verb] + [ing] the object | 31.59 | 33.08 | 29.52 | 33.97 |
>
> ### **UO**
>
> | Template | **HM** | **Full** | **Unseen** | **Seen** |
> | --- | --- | --- | --- | --- |
> | a photo of a person [verb] + [ing] the object (default) | 33.64 | 32.79 | 35.04 | 32.34 |
> | person [verb] the object | 33.82 | 32.99 | 35.21 | 32.54 |
> | an example of the person [verb] + [ing] the object | 33.57 | 32.83 | 34.77 | 32.45 |
>
> ### **UV**
>
> | Template | **HM** | **Full** | **Unseen** | **Seen** |
> | --- | --- | --- | --- | --- |
> | a photo of a person [verb] + [ing] the object (default) | 29.92 | 32.78 | 26.88 | 33.74 |
> | person [verb] the object | 29.74 | 32.89 | 26.46 | 33.94 |
> | an example of the person [verb] + [ing] the object | 29.95 | 33.05 | 26.70 | 34.08 |
>
> Template variation shows minimal performance impact, indicating that our prompt learning framework is robust to sentence structure—even with CLIP’s known phrase encoding bias. This suggests that such bias does not significantly affect our method’s effectiveness.
>
> ---
>
> ### 5. Minor points
>
> - **Computational overhead**: All LLM concepts are generated offline, once per (verb, region) pair. The total size of the concept pool is fixed (10 concepts × 3 regions × 117 verbs). Variance computation is also performed once and cached. Thus, our method adds no runtime cost during inference.
> - **Bias propagation**: We acknowledge that LLM-generated concepts may contain bias (e.g., “woman cooking”). In practice, such cases were rare and did not result in consistent prediction shifts. Future work may incorporate human-in-the-loop filtering or bias-aware prompting.
> - **Failure cases**: We will add qualitative failure examples in the supplementary file, highlighting the generated noisy concepts in the final version.
>
> ---
>
> We again thank the reviewer for the encouraging evaluation and for the insightful suggestions on verb variance robustness and Sparsemax sensitivity. We hope our additional analyses and clarifications help address these concerns.

---

> > ### Author Response · Authors · 2025-08-06
> > **Follow-up on rebuttal response**
> >
> > Dear Reviewer,
> >
> > Thank you for your detailed review. We wanted to kindly follow up and check whether our rebuttal addressed your concerns.
> >
> > If any points remain unclear or if you have further suggestions, we would be happy to provide additional clarification during the discussion period.
> >
> > We sincerely appreciate your time and consideration.
> >
> > Best regards,
> > Authors

---

> > ### Comment · Reviewer_FDPj · 2025-08-08
> >
> > The rebuttal partially clarifies certain issues, but my original evaluation persists after full consideration of all feedback.

---

> ### Author Response · Authors · 2025-08-08
>
> Dear Reviewer,
>
> We sincerely appreciate your thorough and constructive review, as well as your thoughtful evaluation of our work.
> Your detailed feedback has been invaluable in refining our understanding and has provided clear directions for future improvement.
> We are grateful for the time and effort you dedicated to assessing our paper and for recognizing the strengths of our approach.
>
> Best regards,
> Authors

---

### Note · Authors · 2025-08-16

We thank the reviewers for their constructive feedback. Reviewer FDPj noted novelty of combining Visual Diversity-aware Prompts and Region-aware Prompts, citing variance injection and Sparsemax retrieval as promising. Reviewer 2XwT valued our intuition on core challenges in HOI detection. Reviewer cCr1 and 7Lyn acknowledged strong results over SoTA. Below, we summarize additional experiments and clarifications from the rebuttal.

**1. Robustness of group-wise variance (FDPj, cCr1)**

Our group-wise variance modeling remains robust even with reduced semantic support. Removing the top-3 similar verbs per unseen verb reduced unseen mAP by only −0.67, showing generalization to isolated verbs. With 5 samples per verb, the drop was ≤0.2 full mAP, confirming stability under limited data.

**2. Effectiveness of Sparsemax (FDPj, 2XwT)**

Sparsemax outperformed other strategies in UO when cleaner GPT-4o object concepts were used, indicating the earlier drop stemmed from noise in LLaMA concepts rather than Sparsemax:

| Method | Full | Unseen | Seen |
| --- | --- | --- | --- |
| Top-3 | 32.94 | 35.57 | 32.41 |
| Softmax | 32.91 | 35.83 | 32.32 |
| **Sparsemax** | **33.12** | **36.42** | **32.45** |

Also, τ-Sparsemax analysis showed ≤0.4 mAP variation, confirming robustness to thresholding.

**3. Novelty (2XwT, 7Lyn)**

VDRP is novel in: (1) deriving prompt distributions from the visual modality to capture intra-class diversity while preserving alignment after training, (2) augmenting prompts with region concepts filtered by Sparsemax, and (3) jointly addressing intra-class diversity and inter-class entanglement in zero-shot HOI detection.

**4. Hyperparameters and Gaussian perturbation (cCr1)**

Rebuttal experiments clarified each component’s effect: α=0.02 (matching CLIP context initialization) and moderate β stabilized training, while Gaussian perturbation improved unseen mAP by up to +0.4.

**5. Comparison with SoTA (7Lyn)**

VDRP surpasses EZ-HOI in all zero-shot settings with up to +2.5 HM gain, while using ~35% fewer parameters, and outperforms SICHOI in unseen splits despite a lighter backbone.

**6. Robustness to template variation (FDPj, 7Lyn)**

VDRP maintains stable performance when prompt templates change; mAP varied by ≤0.3, showing low sensitivity to CLIP phrase encoding bias.

In closing, we have addressed all major concerns and believe the findings strengthen the paper’s clarity and contribution. We thank the reviewers for their valuable feedback.

---

### Decision · Program_Chairs · 2025-09-17

**Decision:**

Accept (poster)

**Comment:**

The paper introduces a dual-module framework (VDRP) for zero-shot HOI detection, combining visual diversity-aware prompts with region-aware prompts. Reviewers agree on the solid motivation, technical soundness, and strong empirical results, with efficiency noted as a clear strength. While novelty was viewed as incremental by some, others recognized meaningful contributions in the variance injection and dual-module design. Concerns about originality and RAP sensitivity were addressed through additional experiments and clarifications. Overall, the consensus is positive, and the reasons to accept outweigh the reservations. I recommend acceptance.